

# Does the weighting of climate simulations result in a more reasonable quantification of hydrological impacts?

Hui-Min Wang[1], Jie Chen[1*], Chong-Yu Xu[1,2], Hua Chen[1], Shenglian Guo[1], Ping Xie[1], Xiangquan Li[1]

[1]State Key Laboratory of Water Resources and Hydropower Engineering Science, Wuhan University, Wuhan, 430072, China
[2]Department of Geosciences, University of Oslo, Oslo, Norway

*Correspondence to*: Jie Chen (jiechen@whu.edu.cn)

**Abstract:** With the increase in the number of available global climate models (GCMs), pragmatic questions come up when using them to quantify the climate change impacts on hydrology: Is it necessary to weight GCM outputs in the impact studies, and if so, how to weight them? Some weighting methods have been proposed based on the performances of GCM simulations with respect to reproducing the observed climate. However, the process from climate variables to hydrological responses is nonlinear, and thus the assigned weights based on their performances in climate simulations may not be translated to hydrological responses. Assigning weights to GCM outputs based on their ability to represent hydrological simulations is more straightforward. Accordingly, the present study assigns weights to GCM simulations based on their ability to reproduce hydrological characteristics and investigates their influence on the quantification of hydrological impacts. Specifically, eight weighting schemes are used to determine the weights of GCM simulations based on streamflow series simulated by a lumped hydrological model using raw or bias-corrected GCM outputs. The impacts of weighting GCM simulations are investigated in terms of reproducing the observed hydrological regimes for the reference period (1970-1999) and quantifying the uncertainty of hydrological changes for the future period (2070-2099). The results show that when using raw GCM outputs with no bias correction, streamflow-based weights better represent the mean hydrograph and reduce the bias of annual streamflow. However, when applying bias correction to GCM simulations before driving the hydrological model, the climate simulations become rather close to the observed climate, so that compared to equal weighting, the streamflow-based weights do not bring significant differences in the multi-model ensemble mean and uncertainty of hydrological impacts. Since bias correction has been an indispensable procedure in hydrological impact studies, the equal weighting method may still be a viable and conservative choice for the studies of hydrological climate change impacts.



## 1 Introduction

Multi-model ensembles (MMEs) consisting of climate simulations from multiple global climate models (GCMs) have been widely used to quantify future climate change impacts and the corresponding uncertainty (Wilby and Harris, 2006; IPCC, 2013; Chiew et al., 2009; Chen et al., 2011; Tebaldi and Knutti, 2007). The number of climate models has increased rapidly, resulting in the obviously growing size of MMEs. For example, the Coupled Model Inter-comparison Project Phase 5 (CMIP5) archive contains 61 GCMs from 28 modeling institutes, with some GCMs providing multiple simulations (Taylor et al., 2012). Due to the lack of consensus on the proper way to combine the simulations of a MME, the prevailing approach is that of model democracy ("one model one vote") for the sake of simplicity, where each member in an ensemble is considered to have equal ability to simulate historical and future climates. The model democracy method has been applied to many global and regional climate change impact studies (e.g., IPCC, 2014; Minville et al., 2008; Maurer, 2007). Although it has been reported that the equal average of a multi-model ensemble often outperforms any individual model in regards to the reproduction of the mean state of observed historical climate (Gleckler et al., 2008; Reichler and Kim, 2008; Pincus et al., 2008), the use of unequal weights has been recommended in some studies (Xu et al., 2010; Giorgi and Mearns, 2002; Murphy et al., 2004).

Several studies have raised concerns about the strategy of model democracy, due to the following two reasons (Lorenz et al., 2018; Knutti et al., 2017; Cheng and AghaKouchak, 2015). First, GCM simulations in an ensemble do not have identical skills at representing historical climate observations. They may perform differently in simulating future climate. GCM performances may also vary by their variables and locations (Hidalgo and Alfaro, 2015; Su et al., 2013), which further challenges the rationality of model democracy in regional impact studies. Second, equal weights imply that the individual members in an ensemble are independent of each other. However, some climate models share common modules, parts of codes, parameterizations and so on (Knutti et al., 2010; Sanderson et al., 2017). Some pairs of GCMs submitted to the CMIP5 database only differ in the spatial resolution (e.g. MPI-ESM-MR and MPI-ESM-LR; see Giorgetta et al., 2013). The replication or overlapping in these GCMs may lead to the inter-dependence of MMEs, resulting in common biases towards the replicating section and inflating confidence in the projection uncertainty (Sanderson et al., 2015; Jun et al., 2012).

With the intention of improving climate projections and reducing the uncertainty, some weighting approaches have been proposed to assign unequal weights to climate model simulations according to their performances with respect to reproducing some diagnostic metrics of historical climate observations (Murphy et al., 2004; Sanderson et al., 2017; Cheng and AghaKouchak, 2015). For example, Xu et al. (2010) apportioned weights for GCMs based on their biases to the observed data in terms of two diagnostic metrics (climatological mean and inter-annual variability) for producing probabilistic climate projections. Lorenz and Jacob (2010) used errors in the trends of temperature to evaluate climate projections and determine weights. Other criteria have also been introduced into model weighting as a complement to the performance criterion. Some examples are the convergence of climate projections for a future period (Giorgi and Mearns, 2002) and the interdependence among climate models (Sanderson et al., 2017).



Despite the different diagnostic metrics or definitions of model performances employed in these weighting methods, weights are commonly determined with respect to the ability of climate simulations at reproducing observed climate variables, such as temperature and precipitation (e.g., Chen et al., 2017; Wilby and Harris, 2006; Xu et al., 2010). However, for the impact studies, the relationship between climate variables and the impact variable is often not straightforward or explicit. In other words, the process from climate variables to their impacts may not be linear (Wang et al., 2018; Chen et al., 2016; Risbey and Entekhabi, 1996; Whitfield and Cannon, 2000). For example, Mpelasoka and Chiew (2009) reported that in Australia, a small change in annual precipitation can result in a several-times change in annual runoff. Thus, the weights calculated in the climate world may not be effective in the impact field.

In addition, a number of climate variables may determine the climate change impacts on a single environmental sector. For example, the runoff generation in a watershed is usually determined by precipitation, temperature, and other climate variables. Thus, it is not an easy task to determine the relative importance of each climate variable in impact studies, which is the other challenge to combining sets of weights based on different climate variables into a single set of weights for impact simulations. Previous studies have usually assumed that all variables are equally important and had an equal weight assigned to each climate variable (Xu et al., 2010; Chen et al., 2017; Zhao, 2015). However, these climate variables are usually not equally important in the impact field. For example, precipitation may be more important than temperature for a rainfall-dominated watershed, but could be different for a snowfall-dominated watershed. Thus, it may be more straightforward to calculate the weights for GCMs based on their ability to reproduce the single impact variable instead of multiple climate variables. Such a method would integrate the synthetic ability of GCMs in terms of simulating multiple climate variables to that of one impact variable. In addition, this method could also circumvent the previous problem of potential nonlinearity between climate variables and the impact variable.

Accordingly, the objectives of this study are to assign weights to GCM simulations indirectly according to their ability to represent hydrological observations, and to assess the impacts of these weighting methods on the assessment of hydrological responses to climate change. The case study was conducted over two watersheds with different climatic and hydrological characteristics. Seven weighting methods were used to assign unequal weights based on streamflow series simulated using the raw GCM outputs or bias-corrected outputs. The impacts of unequal weights are assessed and compared to the equal weighting method in terms of multi-model ensemble mean and uncertainty related to the choice of a climate model.

## 2 Study area and data

### 2.1 Study Area

This study was conducted over two watersheds with different climate and hydrological characteristics: the rainfall-dominated Xiangjiang watershed and the snowfall-dominated Manicouagan-5 watershed (Figure 1). The Xiangjiang River is one of the largest tributaries of the Yangtze River in central-southern China, and its drainage area is about 94 660 km$^2$ (Figure 1a). A catchment with a surface area of about 52 150 km$^2$ above the Hengyang gauged station was used in this study. The





catchment is heavily influenced by the East Asian Monsoon, which causes a humid subtropical climate with hot and wet summers and mild winters. The average temperature over the catchment is about 17 ℃ with the coldest month averaging about 7 ℃. The average annual precipitation is about 1570 mm, of which 61% falls in the wet season from April to August. The daily averaged streamflow at the Hengyang gauged station is around 1400 $m^3s^{-1}$. The annual average of summer peak streamflow is about 4420 $m^3s^{-1}$, mainly due to summer extreme rainfalls.

The Manicouagan-5 watershed, located in the center of the Province of Quebec, Canada, is the largest sub-basin of the Manicouagan watershed (Figure 1b). Its drainage area is about 24 610 $km^2$, most of which is covered by forest. At the outlet of the Manicouagan-5 River is the Daniel-Johnson Dam, the longest multiple-arch-buttress dam in the world. The Manicouagan-5 watershed has a continental subarctic climate characterized by long and cold winters. The average temperature over the watershed is about -3 ℃, with nearly half of the year having a daily temperature below 0 ℃. The average annual precipitation is about 912 mm, evenly distributed over each year. The average discharge at the outlet of the Manicouagan-5 River is about 530 $m^3s^{-1}$. Snowmelt contributes to the peak discharge during May, whose annual average is about 2200 $m^3s^{-1}$.

## 2.2 Data

This study used daily maximum and minimum temperatures and precipitation from observation and GCM simulations for both watersheds. The observed meteorological data for the Xiangjiang watershed were collected from 97 precipitation gauges and 8 temperature gauges. Streamflow series were collected from the Hengyang gauged station. For the Manicougan-5 watershed, the observed meteorological data were extracted from the gridded dataset of Hutchinson et al. (2009), which is interpolated from daily station data using a thin-plate smoothing spline interpolation algorithm. The hydrological data were collected from the Daniel Johnson Dam using mass balance calculations. All the observation data for both watersheds cover the historical reference period (1970-1999).

For the climate simulations, maximum and minimum temperatures and precipitation of 29 GCMs were extracted from the CMIP5 archive over both watersheds (Table 1). All simulations cover both the historical reference period (1970-1999) and the future projection period (2070-2099). One Representative Concentration Pathway (RCP8.5) was used in terms of climate projections in the future period. RCP8.5 was selected because it projects the most severe increase in greenhouse gas emissions among the four RCPs, and it is often used to design conservative mitigation and adaptation strategies (IPCC, 2014).

## 3 Methodology

To begin the process of calculating the weight for each GCM simulation, a multi-model ensemble constructed by 29 CMIP5 GCMs was utilized to drive a calibrated hydrological model over the two watersheds. Two experiments were designed to generate the ensembles of streamflow simulations. The first experiment drives the hydrological model using raw GCM outputs with no bias correction, while the second drives the hydrological model using bias corrected climate simulations. Although it is not common to use raw GCM simulations for hydrological impact studies, the rationale for using them in this study is to





examine the impacts of bias correction on weighting GCMs. The bias correction may adjust the relative performances between climate simulations and thus affect the determination of the relative weight for each ensemble member. Based on the ensemble of hydrological simulations from GCM outputs, eight weighting methods were employed to determine the weights of each GCM and to combine ensemble members for the assessment of hydrological climate change impacts. More detailed information is given below.

### 3.1 Bias correction

Since the raw outputs of GCMs are often too coarse and biased to be directly input into hydrological models for impact studies, bias correction is commonly applied to GCM outputs prior to the runoff simulation (Wilby and Harris, 2006; Chen et al., 2011; Minville et al., 2008). A distribution-based bias correction method, the daily bias correction (DBC) method of Chen et al. (2013), was used in this study. DBC is the combination of the local intensity scaling (LOCI) method (Schmidli et al., 2006) and the daily translation (DT) method (Mpelasoka and Chiew, 2009). The LOCI method was used to adjust the wet-day frequency of climate model simulated precipitation. A threshold was determined for the reference period to insure that the simulated precipitation occurrence is identical to the observed precipitation occurrence. The same threshold was then used to correct the wet-day frequency for the future period. The DT method was used to correct biases in the frequency distribution of simulated precipitation amounts and temperature. The frequency distribution was represented by 100 percentiles ranging from the 1st to the 100th, and the correction factors were calculated for each percentile. The same correction factors were then employed to correct the distributions for the future period. The use of distribution-based biases facilitates the use of different correction factors for different levels of precipitation. Some studies have shown the advantages of distribution-based bias correction over other correction methods in the assessment of hydrological impacts (Chen et al., 2013; Teutschbein and Seibert, 2012).

### 3.2 Runoff simulation

The runoff was simulated using a lumped conceptual hydrological model, GR4J-6, which couples a snow accumulation and melt module, CemaNeige, with a rainfall-runoff model, GR4J (Arsenault et al., 2015). The CemaNeige model divides the precipitation into liquid and solid according to the daily temperature range, and generates snowmelt depending on the thermal state and water equivalent of the snowpack (Valéry et al., 2014). CemaNeige has two free parameters: the melting rate and the thermal state coefficient. The GR4J model consists of a production reservoir and a routing reservoir (Perrin et al., 2003). A portion of net rainfall (liquid precipitation with evaporation subtracted) goes into the production reservoir, whose leakage forms the effective rainfall when combined with the other proportion of net rainfall. The effective rainfall is then divided into two flow components. Ninety percent of the effective rainfall routes via a unit hydrograph and enters into the routing reservoir. The other 10% generates the direct flow through the other unit hydrograph. There is a groundwater exchange between the direct flow and the outflow nonlinearly generated by the routing reservoir. Four free parameters in GR4J must be calibrated:



the maximum capacity of the production reservoir, the groundwater exchange coefficient, the one-day-head maximum capacity of the routing reservoir and the time base of unit hydrograph.

The time periods of the observed data used for hydrological model calibration and validation are presented in Table 2. The shuffled complex evolution optimization algorithm (Duan et al., 1992) was employed to optimize the parameters of GR4J-6

for both watersheds. The optimized parameters were chosen to maximize the Nash-Sutcliffe Efficiency (NSE) criteria (Nash and Sutcliffe, 1970). The selected sets of parameters yield NSEs greater than 0.87 for both calibration and validation periods, indicating the reasonable performance of GR4J-6 and the high quality of the observed datasets for both watersheds.

### 3.3 Weighting Methods

Raw and bias-corrected climate simulations were input to the calibrated GR4J-6 model to generate raw and bias-corrected

streamflow data series, respectively. Eight weighting methods were then employed to determine the weight of each hydrological simulation, including the equal weighting method (model democracy) and 7 unequal weighting methods. All of the unequal weighting methods are described in detail in the supplementary material so they are only briefly presented herein. Seven unequal weighting methods consist of two multiple criteria-based weighting methods and five performance-based weighting methods. The two multiple criteria-based weighting methods are the reliability ensemble averaging method (REA)

and the performance and interdependence skill (PI). The REA method considers both the bias of a GCM to observation in the reference period (performance criterion) and its similarity to other GCMs in the future projection (convergence criterion) (Giorgi and Mearns, 2002). The PI method weights an ensemble member according to its bias to historical observation (performance criterion) and its distance to other ensemble members in the reference period (interdependence criterion) (Knutti et al., 2017; Sanderson et al., 2017). The biases and distances in the REA and PI methods were calculated based on the

diagnostic metric of the climatological mean.

The five performance-based weighting methods are the climate prediction index (CPI), upgraded reliability ensemble averaging (UREA), the skill score of the representation of the annual cycle (RAC), Bayesian model averaging (BMA), and the evaluation of the probability density function (PDF). All of these methods only consider the differences of climate simulations to historical observation, but they differ in the metrics or algorithms used to determine weights. The CPI assigns weights based

on the biases in the climatological mean and assumes that the simulated climatological mean follows a Gaussian distribution (Murphy et al., 2004). UREA considers biases in both the climatological mean and the inter-annual variance to determine weights (Xu et al., 2010). Both the RAC and BMA calculate weights based on monthly series. The RAC defines a skill score in simulating the annual cycle according to the relationship among the correlation coefficient, standard deviations and centered root-mean-square error (Taylor, 2001). BMA combines the results of multiple models through the Bayesian theory (Duan et

al., 2007; Raftery et al., 2005; Min et al., 2007). The PDF determines weights according to the overlapping area of probability density function between daily simulations and observations (Perkins et al., 2007).





Using all eight methods, the weights were respectively calculated for streamflow data series simulated by raw GCM outputs and bias-corrected outputs. For a comparison, the above methods were also used to calculate weights based on raw or bias-corrected temperature and precipitation series in terms of performances in simulating observed temperature and precipitation.

### 3.4 Data Analysis

The extent of inequality of each set of weights was first investigated by the entropy of weights (Déqué and Somot, 2010). The entropy of weights reflects the extent of how a weighting method discriminates the relative reliability between GCM simulations. Next, in order to investigate the impacts of weighting GCM simulations for hydrological impact studies, unequal weights were used to combine the ensemble of hydrological simulations. The impacts of unequal weights were compared to the results obtained using the equal weighting method. The comparison focuses on three aspects: (1) the simulation of reference and future hydrological regimes; (2) the bias of the multi-model ensemble mean during the reference period; and (3) the uncertainty of changes in hydrological indices between future and reference periods.

Specifically, when using the entropy of weights (Eq. (1)), the entropy reaches a maximum value when the weights are equally distributed among ensemble members. A smaller entropy indicates a larger difference among the weights of ensemble members. Thus, the entropy reflects the extent of inequality for a set of weights:

$$E = -\sum_{i=1}^{N} w_i \ln w_i \tag{1}$$

where $w_i$ is the weight assigned to the $i$th ensemble member, and $N$ is the total number of ensemble members.

Since weighting methods are usually proposed to reduce biases in the ensemble of climate simulations, the multi-model ensemble means determined by these weights are then evaluated in terms of the representation of observation during the reference period. The multi-year averages of three hydrological indices were calculated for each streamflow simulation:(1) annual streamflow; (2) peak streamflow; and (3) the center of timing of annual flow (tCMD: the occurrence day of the midpoint of annual flow). The multi-model mean indices were then obtained based on the weights assigned to each simulation and compared to the indices of observation.

The influences of weighting on the uncertainty of hydrological impacts related to the choice of GCMs are investigated in terms of the changes in four hydrological indices between the reference and future periods: (1) mean annual streamflow; (2) mean streamflow during the high flow period; (3) mean streamflow during the low flow period; and (4) mean peak streamflow (the periods of high and low flow are shown in Table 2). The Monte-Carlo approach was introduced to sample the uncertainty for unequally weighted ensembles (Wilby and Harris, 2006; Chen et al., 2017). The hydrological indices were randomly sampled one thousand times based on the calculated weights. For example, if a climate model simulation is assigned a weight of 0.2, the hydrological index simulated by that climate simulation has a probability of 20% to be chosen as the sample in each Monte-Carlo experiment.



## 4 Results

### 4.1 Weights of GCMs

Figure 2 presents the weights calculated based on the streamflow data series simulated by raw GCM outputs and bias-corrected outputs for 8 (one equal and 7 unequal) weighting methods over two watersheds. These results show the ability of
5 different weighting methods to distinguish the performance or reliability of individual ensemble members. The entropy of weights was also calculated to quantify the extent of this disproportion for each set of weights (Table 3). Some weighting methods tend to aggressively discriminate the reliability of GCMs and assign differentiated weights to ensemble members, while other methods assign similar weights to each of them. Specifically, when calculating weights based on raw GCM-simulated streamflows, REA, UREA and CPI produce the weights that most radically discriminate ensemble members among
10 all eight weighting methods for both watersheds. The RAC method generates less differentiated unequal weights, followed by the BMA and PI methods, but weights assigned by the PDF method closely resemble the equal weighting method. However, when calculating weights based on bias-corrected GCM-simulated streamflows, the inequality of weights is reduced, and all the unequal weighting methods receive a lower entropy of weights for both watersheds (Table 3). Most sets of these weights become similar to the equal weighting method, with the exception of REA and UREA for the Xiangjiang watershed, and REA
for the Manicouagan-5 watershed (Fig. 2). This result was expected, as the bias correction method brings all GCM simulations to be close to the observations. The differences among GCM simulations become greatly reduced.

In addition, the weights based on the raw and bias-corrected temperature and precipitation time series of GCM simulations were also calculated and are shown in Fig. S1. For the weights based on the raw temperature and precipitation, REA, UREA and CPI still generate the most unequal weights among these weighting methods over both watersheds, as Table 3 indicates.
Again, the weights become equalized when calculating weights based on bias-corrected temperature and precipitation.

### 4.2 Impacts on the hydrological regime

The weights determined by eight weighting methods were first utilized to combine GCM-simulated streamflow series. Figure 3 shows the weighted multi-model mean of monthly mean streamflow for the Xiangjiang watershed. The gray envelope represents the range of monthly mean streamflow simulated using 29 GCM simulations. At the reference period, streamflows
simulated by raw GCMs cover a wide range (Figure 3a). However, the equal-weighted multi-model mean streamflow performs better than most of the streamflow series simulated by individual GCMs with respect to reproducing the observed streamflow; even so, the equal-weighted ensemble mean still underestimates the streamflow before the peak (January – May) and overestimates it after the peak (June – September).

For the ensemble mean combined by unequal weights, the three weighting methods that generate highly differentiated
weights (REA, UREA and CPI) outperform the equal weighting method with respect to reproducing the observed monthly mean streamflow. The BMA and RAC methods improve the performance of streamflow simulations before the peak at the cost of performance after the peak, while an opposite pattern is observed when using the PI method. The PDF method generates



an ensemble mean of monthly mean streamflows almost identical to that of the equal weighting method. This is an expected result, as the PDF method assigns almost identical weights to all GCM simulations.

Weights calculated based on the raw temperature and precipitation of GCM outputs were also used to construct the ensemble mean of monthly mean streamflows (Fig. S2a,b). Particularly, the ensemble mean hydrographs combined using the

REA, UREA and CPI methods largely deviate from the observation. Although REA, UREA and CPI generate highly differentiated weights when based on GCM raw temperatures, their generated ensemble mean streamflows are significantly inferior to that generated by equal weights (Fig. S2a). In addition, when using raw precipitation to calculate weights, the weighting methods perform worse than or similar to those calculated based on streamflow series (Fig. S2b). This reflects the advantage of weighting streamflow series in terms of reproducing the observed mean hydrograph.

The bias correction method can reduce the biases of precipitation and temperature in representing the mean monthly streamflow for the reference period, as indicated by the narrowed envelope (Figure 3c), although a small amount of uncertainty is still observed. The reduction of biases brings about similar weights for all GCM-simulated time series when using bias-corrected GCM-simulated streamflows. Thus, the multi-model ensemble means of monthly mean streamflow constructed by all unequal weighting method are very similar to those constructed by the equal weighting method, as shown in Figure 3c.

For the bias-corrected GCM-simulated streamflow at the future period (Figure 3d), a larger uncertainty related to the use of climate models is observed, as indicated by the wider envelope of the mean monthly streamflow. This may be because the bias of GCM outputs is non-stationary. All bias correction methods are based on a common assumption that the bias of climate model outputs is constant over time. However, this assumption may not be true, due to natural climate variability and climate sensitivity (Hui et al., 2018). In addition, if the bias of climate model outputs is not stationary, it is unnecessary to use unequal

weighting methods. In other words, the bias non-stationarity implies that climate models differ in their ability to simulate precipitation and temperatures for the future period. The weights calculated at the reference period may not be applicable at the future period. The results of this study also proved this conclusion, as all of the weighting methods project similar ensemble means of mean monthly streamflows for the future period.

Figure 4 presents the same information as Figure 3 but for the Manicouagan-5 watershed. Nearly half of the monthly mean

streamflow time series simulated by raw GCM outputs have delayed peak (June) compared to the observed one (May) at the reference period, which leads to the delayed peak streamflow of the weighted multi-model mean streamflows for all weighting methods (Figure 4a). Nonetheless, when using raw GCM-simulated streamflow series to calculate weights, the multi-model mean streamflows perform better than or similar to those simulated using GCM raw temperature and precipitation data (Fig. S2c). However, for the bias-corrected streamflow series, the uncertainty of monthly streamflows simulated by individual bias-

corrected GCMs is largely reduced and the problem of delayed peak streamflow is corrected (Figure 4c). Similar to the case in the Xiangjiang watershed, all unequally weighted multi-model mean streamflows are identical to that of the equal weighting method. For the future period, although the uncertainty of single bias-corrected GCM-simulated streamflows increases (Figure 4d), there are still very little differences among the future multi-model mean streamflows combined by different weighting methods.



### 4.3 Bias in multi-model mean

In order to quantify the performance of weighting methods with respect to reproducing the multi-model ensemble mean, biases of the multi-model ensemble mean relative to corresponding observation were calculated for the reference period in terms of three hydrological indices (mean annual streamflow, mean peak streamflow and mean center of timing of annual flow; tCMD). A smaller bias represents a better performance. Figure 5 presents the biases of weighted multi-model mean indices over the Xiangjiang watershed. For the streamflows simulated using raw GCM outputs, the weighting methods show varied performance in terms of reproducing observed indices (Figure 5a-c). Except for the PI method, the unequal-weighted multi-model means more or less outperform the equal weighting method in terms of reducing biases in mean annual streamflow and mean center timing, while an opposite result is observed in mean peak streamflow. This may because most weighting methods only consider the mean value (climatological mean or monthly mean series) when evaluating GCMs, and none of them include peak or extreme values. Additionally, weights calculated based on the raw temperature and precipitation of GCM outputs were used to calculate multi-model mean indices for comparison (Fig. S3a-c). When using raw temperature series of GCMs to determine weights, they often bring about more biases in mean annual streamflow and tCMD. The weights based on raw precipitation show some superiority in reducing bias in mean peak streamflow. However, when using bias-corrected GCM-simulated streamflows to calculate weights (Figure 5d-f), the biases in multi-model mean indices are much less varied among different weighting methods. This is similar to the previous results of hydrological regimes.

For the case in the Manicouagan-5 watershed, twenty-five of the 29 streamflow series simulated by raw GCMs have larger mean annual streamflows and mean peak streamflows than those of the observations, and 26 series generate delayed tCMD. This leads to the overestimation of multi-model mean indices for all weighting methods (Figure 6a-c). Compared to the equal weighting method, all unequal weighting methods overcome this overestimation more or less. The three weighting methods that generate highly differentiated weights (REA, UREA and CPI) notably reduce biases for all three hydrological indices. For most weights calculated based on raw temperature and precipitation of GCM outputs (Fig. S3d-f), a certain improvement on mean indices was also observed (the only exception is raw precipitation-based PDF weights). Compared to weights calculated using streamflow series, nearly all weights based on GCM-simulated streamflows reduce more biases than those based on temperature and precipitation. However, when using bias-corrected GCM-simulated streamflows (Fig. 6d-f), again, all weighting methods generate very similar mean indices to the equal weighting method, since the biases among different GCM-simulated streamflows have been largely reduced by the bias correction method.

### 4.4 Impacts on uncertainty

In addition to the multi-model ensemble mean, the impacts of weighting GCM simulations on uncertainty of hydrological responses also need to be assessed. Thus, this study also evaluated how unequal weighting methods affect the uncertainty of hydrological impacts related to the choice of GCMs. Figures 7 and 8 present the box plots of changes in 4 hydrological indices (mean annual streamflow, mean streamflow during the high/low flow periods and mean peak streamflow) between the



reference and future periods. The box plots of the equal weighting method are depicted using 29 values simulated by each climate simulation, while the box plots of 7 unequal weighting methods are constructed using 1,000 values sampled by the Monte-Carlo approach based on assigned weights. For example, a simulation with 2-times the weight as another one will occur 2-times as often as that one in the 1,000 samples of Monte-Carlo experiments. While the 1,000 samples still only consist of

the 29 values, the occurrence of each value reflects its possibility to be chosen and presents the uncertainty related to the choice of GCMs determined by assigned weights.

Figure 7 presents the uncertainty of hydrological changes for the Xiangjiang watershed. When using raw GCM-simulated streamflows (Figure 7a-d), depending on the weighting methods, unequal weights show the varying effects on the uncertainty. Both the PDF and PI methods suggest similar uncertainties to those of the equal weighting method for all four hydrological

indices. The BMA and RAC methods generate slightly larger uncertainty for the change in mean annual streamflow and slightly smaller uncertainty of the change in low streamflow. The two weighting methods that generate the most differentiated weights (REA and UREA) largely reduce the uncertainty and increase the changes of the upper and lower probabilities for all four hydrological variables. The impacts of weights calculated based on raw GCM temperature and precipitation series were also analyzed (Fig. S4a-d). When calculating weights based on raw temperature, REA, UREA and CPI tend to aggressively reduce

the uncertainty in mean high streamflow and peak streamflow. Precipitation-based weights show similar influences on uncertainty as weights based on streamflows. However, for the bias-corrected GCM-simulated streamflows (Figure 7e-h), the uncertainty of changes in the four hydrological indices is similar among all weighting methods.

Figure 8 presents the uncertainty of hydrological impacts in terms of four hydrological indices over the Manicouagan-5 watershed. For weights calculated using raw GCM-simulated streamflows (Figure 8a-d), only UREA clearly reduces the

uncertainty for mean annual streamflow. The REA, UREA and CPI methods reduce the uncertainty for mean low streamflow and decrease its value of upper probability. There are few differences in the uncertainty of mean high streamflow and peak streamflow among all weighting methods. However, when using bias-corrected GCM-simulated streamflows (Figure 8e-h), again, the uncertainty of changes in all four hydrological indices is very similar among most of the weighting methods. Only CPI suggests slight increases in changes of the lower probability.

**5 Discussion**

In addition to the equal weighting method, which is a normal strategy for handling multi-model ensembles, many studies have also proposed various unequal weighting methods for impact studies (e.g., Giorgi and Mearns, 2002; Sanderson et al., 2017; Xu et al., 2010; Min et al., 2007; Murphy et al., 2004). Most of these methods calculate weights based on the reliability of GCM simulations relative to observed climates, or at least adopt their reliability as one of their weighting criteria. In other

words, the performances of GCM simulations are usually evaluated by comparing them to observed climate using certain metrics. However, this method may have two problems. First, the trade-off between multiple climate variables related to the impact variable remains uncertain, which leads to difficulty in obtaining a single set of weights for impact studies. Second, the





relationship between climate variables and the impact variable is often non-linear and not explicit, which may jeopardize the validity and reasonableness of climate-based weights in the impact studies. Some examples are the weights based on raw GCM temperature in the Xiangjiang watershed, which lead to obviously biased multi-model mean hydrographs at the reference period. However, using the weights calculated based on raw GCM precipitation does not lead to such biases. This may be

because the runoff generation in the Xiangjiang watershed is dominated more by rainfall than temperature. Therefore, weights calculated using temperature may not reflect a GCMs' reliability that is relevant to hydrological responses. On the contrary, for the snow-dominated Manicouagan-5 watershed, the snowmelt-driven spring flood is an important characteristic of its hydrological regime, and both temperature and precipitation conditions have large influences on this process. Thus, weights based on temperature and precipitation do not lead to obviously biased multi-model mean hydrographs. Furthermore, over

both watersheds, most weights calculated using raw GCM-simulated streamflows reduce more biases of the mean annual streamflow than those based on raw temperature and precipitation. This is as expected, because weights based on streamflows directly reflect how GCM simulations conform to the observed streamflow and are not affected by the non-linear relationship between climate variables and impact variables. Generally, weights calculated based on streamflows not only circumvent the above two problems, they also bring about fewer biases in mean annual streamflow for the multi-model means.

Since bias correction methods are routinely applied to GCM outputs for hydrological impact assessments of climate change, this study considered two experiments where raw and bias-corrected GCM-simulated streamflows were used separately to determine weights. The weights were correspondingly assigned to two types of streamflows and their impacts on hydrological responses to climate change were compared. As shown in Figures 3 and 4, biases in the simulated mean monthly streamflows are greatly reduced for the reference period. This reduction in biases directly affects the determination of weights. When using

bias-corrected GCM simulations, all of the weighting methods assign similar weights to ensemble members. Furthermore, two experiments revealed different performances of unequal weights in quantifying hydrological impacts. For the experiment with raw GCM-simulated streamflows, the impacts of unequal weighting vary with the choice of weighting methods. With bias-corrected GCM-simulated streamflows, the results are totally different than in the first experiment. Not only are all the weighted multi-model means of monthly mean streamflows similar to those of the equal weighting method, the uncertainty of

the hydrological impacts is also similar among all of the weighting methods. This is because performance-related weighting methods assign similar weights to all simulations. Since bias correction has been an indispensable procedure for hydrological impact studies, and unequal weighting methods do not have a large influence on impact results, the model democracy approach is still recommended in dealing with multi-model ensembles.

Despite the choices of variables used to calculate weights, the establishment of any weighting method involves subjective

choices of diagnostic metrics, its translation to performance measurement, and normalization to weights (Knutti et al., 2017; Santer et al., 2009). For example, in the RAC method, the correlation coefficient and standard deviation are used as diagnostic metrics, and GCM skills are measured through the translation of a fourth-order formulation. The skill scores are then divided by their sum to be normalized. Any of these steps can ultimately affect the property of a weighting method. For example, the REA, UREA and CPI methods are inclined to generate more differentiated weights, while other methods assign more similar



weights to ensemble members. However, all of these aspects in weighting methods are often arbitrary or based on expert experience and, thus, can actually introduce several layers of subjective uncertainty. An improper weighting method may even cause a risk of reducing projection accuracy (Weigel et al., 2010), and extremely aggressive weighting may conceal the uncertainty rather than reduce it (Chen et al., 2017).

Moreover, some risks may exist in the usage of weighting methods. Firstly, weights are generally assigned to climate simulations in a static way (i.e. weights in the reference period are the same as those in the future period). This approach is based on the assumption that the performances of GCM simulations are stable and stationary. However, some studies have shown that model skills are nonstationary in a changing climate (Weigel et al., 2010; Miao et al., 2016), and models with better performance in the reference period do not necessarily provide more realistic signals of climate change (Reifen and Toumi,

2009; Knutti et al., 2010). Therefore, the assumption of stationary GCM performances may be questionable. Secondly, performance measurement in most weighting methods only depends on one diagnostic metric, such as the long-term mean state (e.g., 30-year climatological mean in REA, PI, UREA, and CPI methods). It is not clear whether reducing the bias of one specific metric can transfer to other metrics. The weights calculated using the raw GCM-simulated streamflows in the Xiangjiang watershed are one example, where the bias in mean annual streamflow is reduced while the bias in the mean peak

streamflow is enlarged. Some studies have also shown similar problems (Jun et al., 2012; Santer et al., 2009). For example, Jun et al. (2012) demonstrated that there is little relationship between a GCMs' ability to reproduce mean temperature state and trend of temperature. Overall, notwithstanding the potential gains obtained by weights calculated using streamflows, the equal weighting method remains an easily available and conservative way to handle the ensemble of multiple climate models for hydrological impact studies.

## 20  6 Conclusion

In order to weight climate models based on runoff simulation and to quantify its influence on hydrological impact assessment, an ensemble of 29 CMIP5 GCMs were weighted by a group of weighting methods based on the their simulated streamflows time series. Raw and bias-corrected GCM simulations were used to drive hydrological models to obtain hydrological simulations. Using streamflows to determine weights is straightforward and can avoid the difficulty of combining

weights based on multiple climate variables for impact studies. The influences of these unequal weights on the assessment of hydrological impacts were then investigated and compared to the common strategy of model democracy.

This study concludes that for the streamflows simulated using raw GCM outputs without bias correction, using unequal weights has some advantages over the equal weighting method in simulating observed hydrographs and in reducing the biases of multi-model means in mean annual streamflow. However, when using bias-corrected GCM outputs to simulate streamflow,

GCM simulations were brought close to the observations by the bias correction method. The weights assigned to climate simulations consequently become similar to each other, resulting in similar multi-model means or uncertainty of hydrological impacts for all of the unequal weighting methods.





Since bias-correction or downscaling has been an indispensable procedure when assessing climate change impacts on hydrology, the equal weighting method is still recommended, or at least, the equal weighting results should be provided to end users along with the unequal weighting results, as well as a detailed explanation of the weighting procedure.

**Acknowledgements**

This work was partially supported by the National Natural Science Foundation of China (Grant No. 51779176, 51539009, 91547205), the Overseas Expertise Introduction Project for Discipline Innovation (111 Project) funded by Ministry of Education and State Administration of Foreign Experts Affairs P.R. China (Grant No. B18037), the Thousand Youth Talents Plan from the Organization Department of CCP Central Committee (Wuhan University, China) and the Research Council of Norway (FRINATEK Project 274310). The authors would like to acknowledge the World Climate Research Program Working Group on Coupled Modelling, and all climate modeling institutions listed in Table 1 for making GCM outputs available. We also thank Hydro-Québec and the Changjiang Water Resources Commission for providing observation data in the Manicouagan-5 and Xiangjiang watersheds, respectively.

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

**Figure 1.** Locations of the (a) Xiangjiang and (b) Manicouagan-5 watersheds. (The study area in the Xiangjiang watershed is one of its sub-basins as the green boundary.)



**Table 1. Information about the 29 GCMs used.**

| No. | Model name | Resolution (Lon. × Lat.) | Institution |
|---|---|---|---|
| 1 | ACCESS1.0 | 1.875 × 1.25 | Commonwealth Scientific and Industrial Research Organization |
| 2 | ACCESS1.3 | 1.875 × 1.25 | (CSIRO) and Bureau of Meteorology (BOM), Australia |
| 3 | BCC-CSM1.1 | 2.8 × 2.8 | Beijing Climate Center, China Meteorological Administration |
| 4 | BCC-CSM1.1(m) | 1.125 × 1.125 | |
| 5 | BNU-ESM | 2.8 × 2.8 | College of Global Change and Earth System Science, Beijing Normal University |
| 6 | CanESM2 | 2.8 × 2.8 | Canadian Centre for Climate Modelling and Analysis |
| 7 | CCSM4 | 1.25 × 0.94 | US National Centre for Atmospheric Research |
| 8 | CESM1(CAM5) | 1.25 × 0.94 | National Science Foundation, Department of Energy, NCAR, USA |
| 9 | CMCC-CMS | 1.875 × 1.875 | Centro Euro-Mediterraneo per I Cambiamenti Climatici |
| 10 | CMCC-CM | 0.75 × 0.75 | |
| 11 | CMCC-CESM | 3.75 × 3.7 | |
| 12 | CNRM-CM5 | 1.4 × 1.4 | Centre National de Recherches Météorologiques and Centre Européen de Recherche et Formation Avancée en Calcul Scientifique |
| 13 | CSIRO-Mk3.6.0 | 1.8 × 1.8 | Commonwealth Scientific and Industrial Research Organization and Queensland Climate Change Centre of Excellence |
| 14 | FGOALS-g2 | 1.875 × 1.25 | LASG, Institute of Atmospheric Physics, Chinese Academy of Sciences, and CESS, Tsinghua University |
| 15 | GFDL-CM3 | 2.5 × 2.0 | NOAA Geophysical Fluid Dynamics Laboratory |
| 16 | GFDL-ESM2G | 2.5 × 2.0 | |
| 17 | GFDL-ESM2M | 2.5 × 2.0 | |
| 18 | INM-CM4 | 2.0 × 1.5 | Russian Institute for Numerical Mathematics |
| 19 | IPSL-CM5A-LR | 3.75 × 1.9 | Institut Pierre Simon Laplace |
| 20 | IPSL-CM5A-MR | 2.5 × 1.25 | |
| 21 | IPSL-CM5B-LR | 3.75 × 1.9 | |
| 22 | MIROC-ESM-CHEM | 2.8 × 2.8 | Japan Agency for Marine-Earth Science and Technology, Atmosphere and Ocean Research Institute (The University of Tokyo), and National Institute for Environmental Studies |
| 23 | MIROC-ESM | 2.8 × 2.8 | |
| 24 | MIROC5 | 1.4 × 1.4 | Atmosphere and Ocean Research Institute (The University of Tokyo), National Institute for Environmental Studies, and Japan Agency for Marine-Earth Science and Technology |
| 25 | MPI-ESM-LR | 1.875 × 1.875 | Max Planck Institute for Meteorology |
| 26 | MPI-ESM-MR | 1.875 × 1.875 | |
| 27 | MRI-ESM1 | 1.125 × 1.125 | Meteorological Research Institute |
| 28 | MRI-CGCM3 | 1.1 × 1.1 | |
| 29 | NorESM1-M | 2.5 × 1.875 | Norwegian Climate Centre |



**Table 2. Nash-Sutcliffe Efficiency (NSE) of hydrological models in the calibration and validation periods.**

| Country | Watershed name | Area (km²) | High flow | Low flow | Calibration period | NSE calibration | Validation period | NSE validation |
|---------|----------------|------------|-----------|----------|--------------------|-----------------| ------------------|----------------|
| China | Xiangjiang | 52150 | Apr-Jun | Jul-Nov | 1975-1987 | 0.912 | 1988-2000 | 0.871 |
| Canada | Manicouagan-5 | 24610 | Mar-Jul | Aug-Feb | 1970-1979 | 0.926 | 1980-1989 | 0.881 |





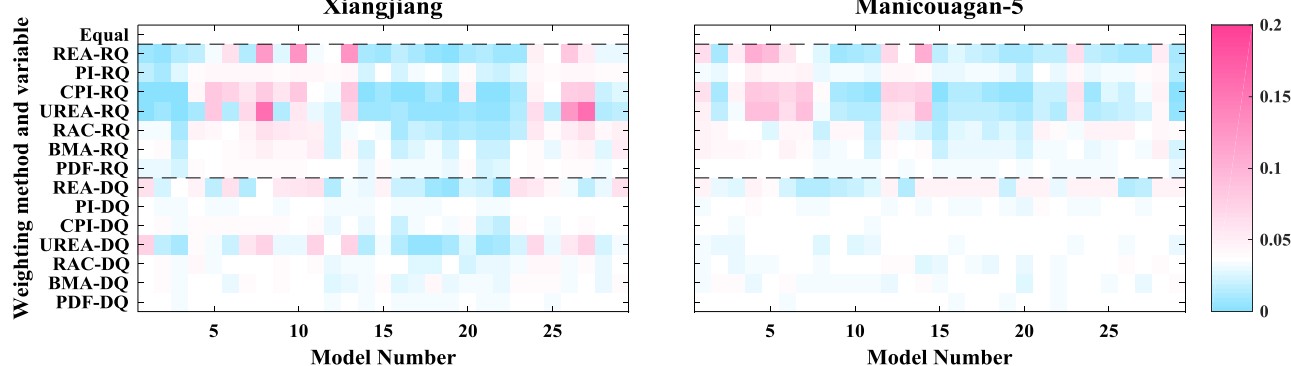

**Figure 2. Weights assigned by equal weighting and 7 unequal weighting methods based on raw climate model data-simulated streamflow (RQ) and bias corrected data-simulated streamflow (DQ) for two watersheds. (Equal weight is presented in white, weights greater than equal are presented in red, and weights less than equal in blue.)**





**Table 3. The entropy of weights calculated by equal weighting and 7 unequal weighting methods based on raw climate model data-simulated streamflow (RQ) and bias corrected data-simulated streamflow (DQ) for two watersheds. The entropy of weights calculated based on raw and bias-corrected temperature (RT and DT) and precipitation (RP and DP) are also presented for comparison.**

| | Xiangjiang watershed | | | | | | Manicouagan-5 watershed | | | | | |
|---|---|---|---|---|---|---|---|---|---|---|---|---|
| | RT | RP | RQ | DT | DP | DQ | RT | RP | RQ | DT | DP | DQ |
| REA | 2.45 | 3.04 | 2.93 | 3.05 | 3.18 | 3.22 | 2.87 | 3.11 | 3.06 | 3.12 | 3.30 | 3.29 |
| PI | 3.34 | 3.35 | 3.33 | 3.37 | 3.37 | 3.37 | 3.34 | 3.34 | 3.34 | 3.36 | 3.36 | 3.37 |
| CPI | 2.46 | 2.92 | 2.86 | 3.37 | 3.36 | 3.35 | 2.99 | 3.12 | 3.00 | 3.37 | 3.37 | 3.37 |
| UREA | 2.72 | 3.00 | 2.73 | 3.33 | 3.22 | 3.15 | 3.02 | 3.15 | 3.10 | 3.33 | 3.35 | 3.36 |
| RAC | 3.37 | 3.35 | 3.25 | 3.37 | 3.36 | 3.36 | 3.37 | 3.36 | 3.32 | 3.37 | 3.36 | 3.36 |
| BMA | 3.34 | 3.36 | 3.33 | 3.36 | 3.36 | 3.36 | 3.35 | 3.36 | 3.35 | 3.37 | 3.36 | 3.36 |
| PDF | 3.36 | 3.37 | 3.36 | 3.37 | 3.37 | 3.37 | 3.37 | 3.37 | 3.37 | 3.37 | 3.37 | 3.37 |
| Equal | 3.37 | | | | | | 3.37 | | | | | |




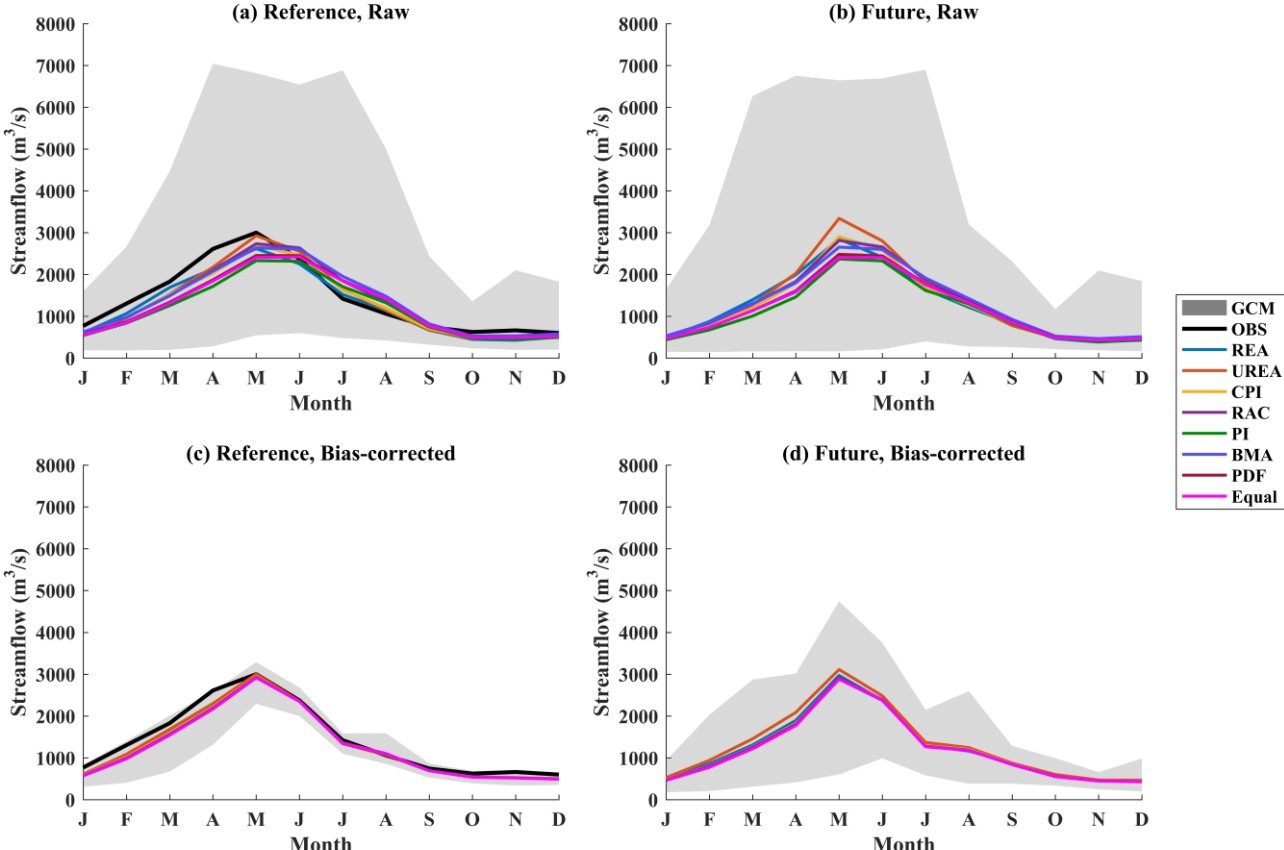

**Figure 3. The envelope of monthly mean streamflows simulated by 29 raw and bias-corrected GCM outputs and the multi-model ensemble means of monthly mean streamflows weighted by 8 weighting methods based on GCM-simulated streamflows over the Xiangjiang watershed for the reference and future periods (OBS = the hydrograph simulated from meteorological observation).**





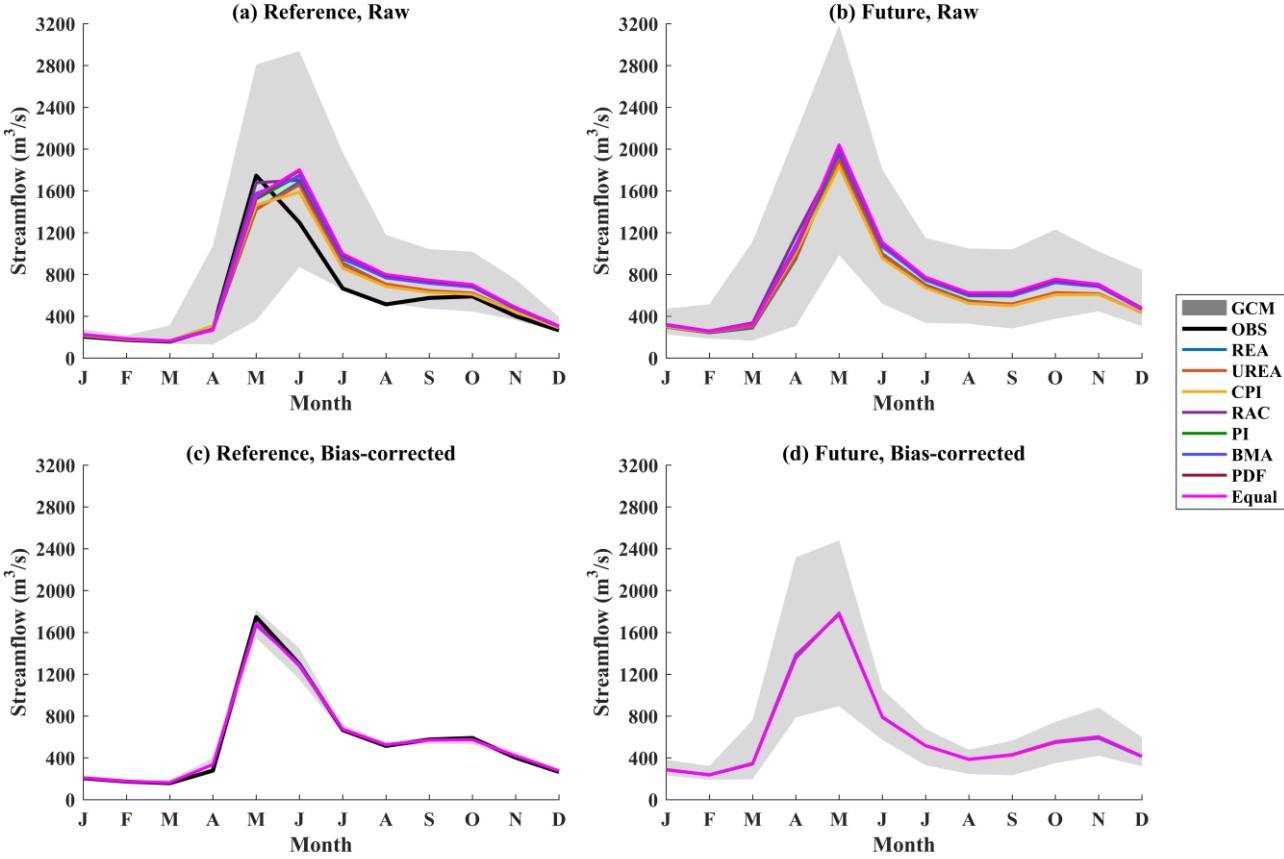

**Figure 4. The envelope of monthly mean streamflows simulated by 29 raw and bias-corrected GCM outputs and the multi-model ensemble means of monthly mean streamflows weighted by 8 weighting methods based on GCM-simulated streamflows over the Manicouagan-5 watershed for the reference and future periods (OBS = the hydrograph simulated from meteorological observation).**



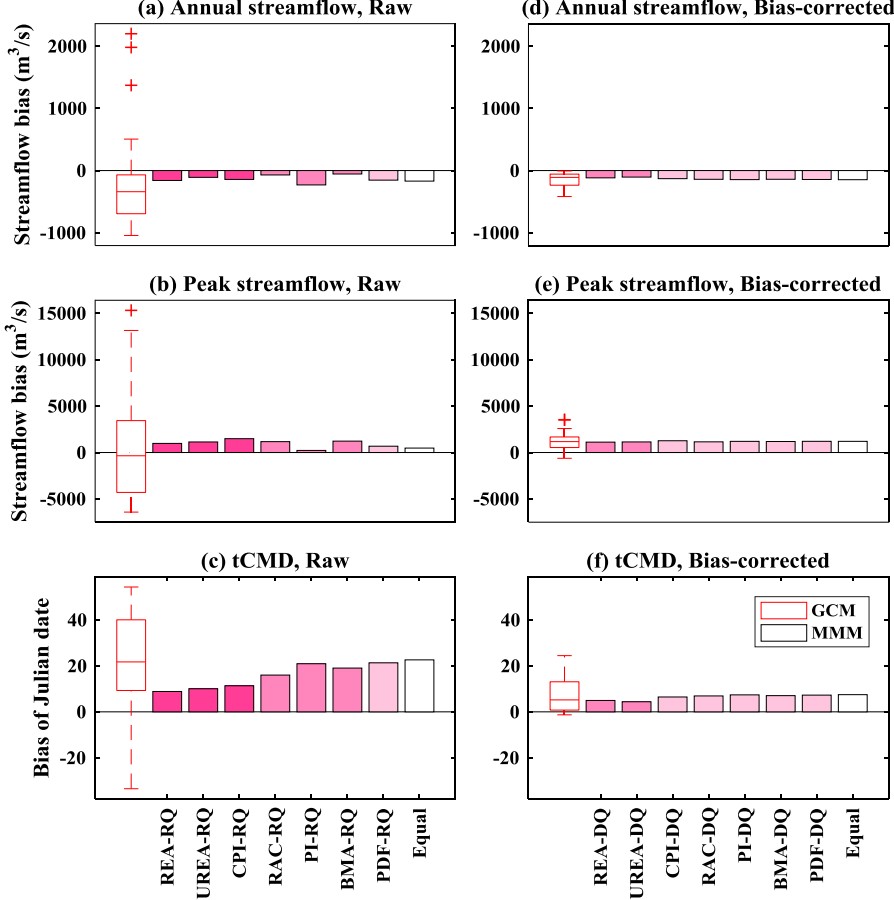

**Figure 5. Bias in mean annual streamflow, mean peak streamflow and mean center of timing of annual flow (tCMD) simulated using 29 raw or bias-corrected GCM outputs and the multi-model means (MMM) combined by weights based on raw (RQ) and bias-corrected (DQ) GCM-simulated streamflows in the Xiangjiang watershed in the reference period. (The depth of pink in the MMM bars represents the level of inequality of weights, as indicated in Table 3.)**



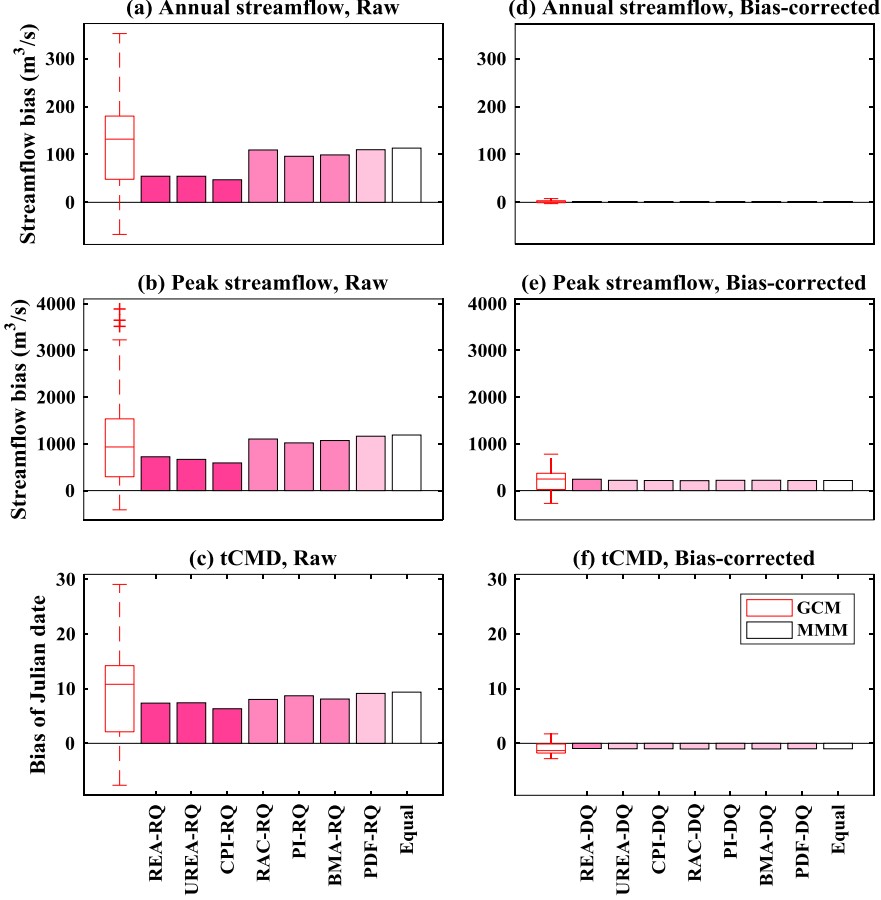

**Figure 6. Bias in mean annual streamflow, mean peak streamflow and mean center of timing of annual flow (tCMD) of streamflows simulated using 29 raw or bias-corrected GCMs and the multi-model means (MMM) combined by weights based on raw streamflows (RQ) and bias-corrected streamflows (DQ) in the reference period in the Manicouagan-5 watershed. (The depth of pink in the MMM bars represents the level of inequality for the corresponding set of weights.)**





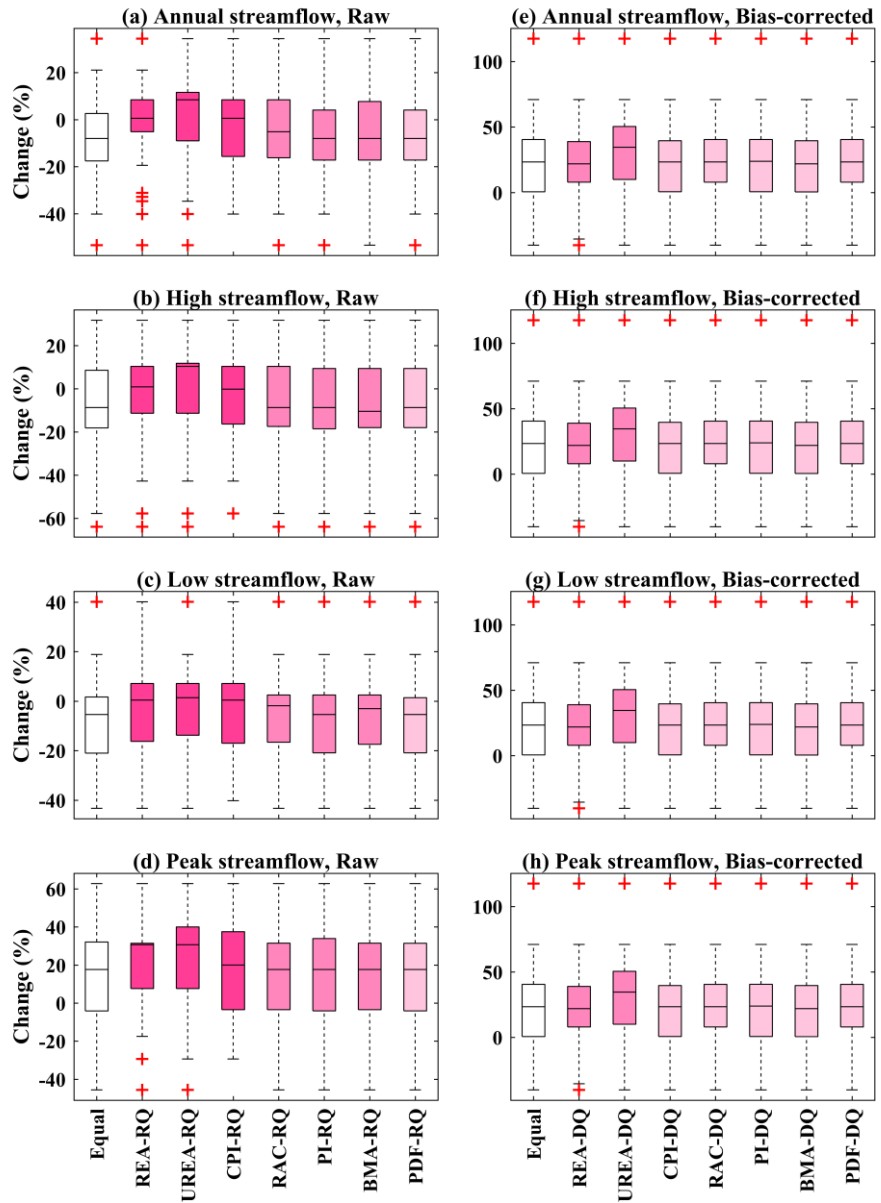

**Figure 7. Box plot of changes in four hydrological indices calculated by raw or bias-corrected GCM-simulated streamflows in the Xiangjiang watershed. The changes of hydrological variables were sampled through the Monte-Carlo approach based on the weights calculated using raw (RQ) or bias-corrected (DQ) GCM-simulated streamflows. (The depth of pink represents the level of inequality of the weights.)**





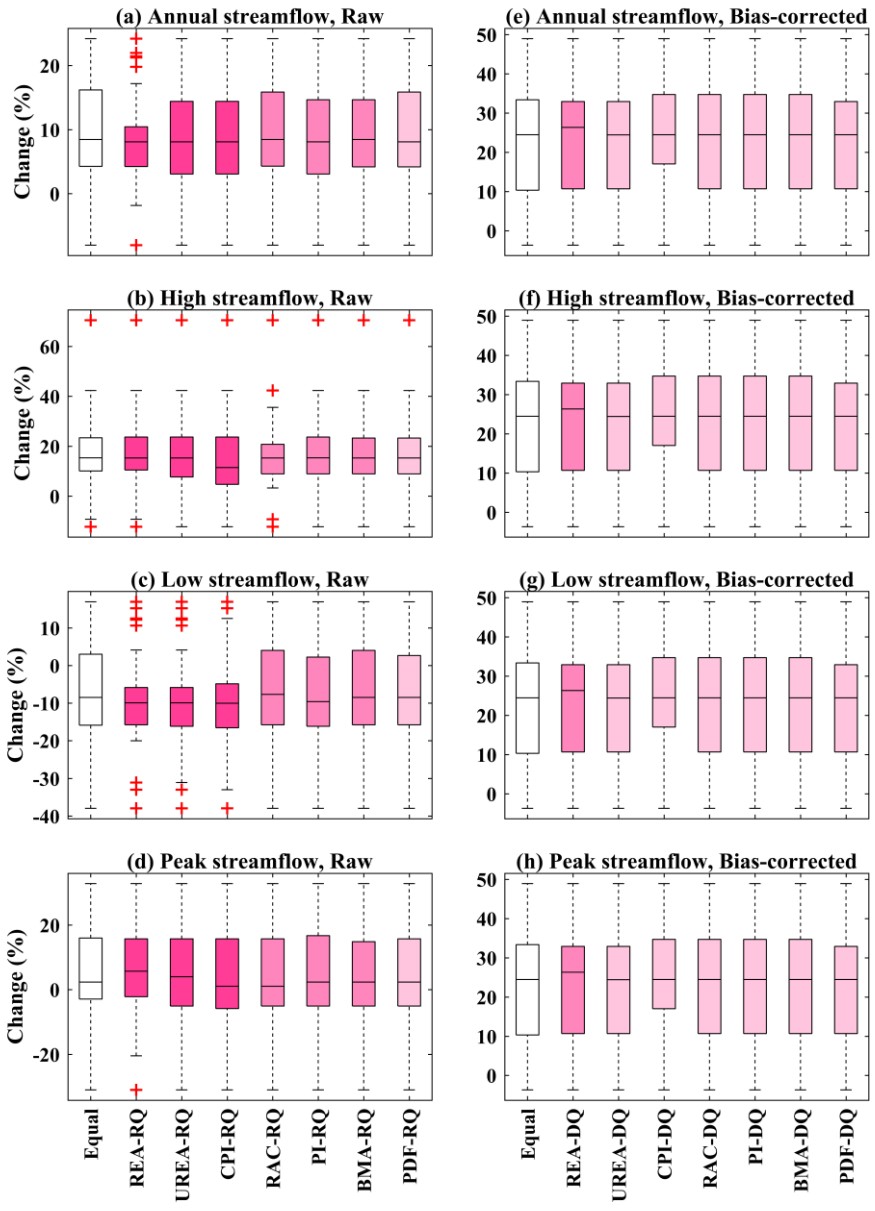

**Figure 8. Box plot of changes in four hydrological indices calculated by raw or bias-corrected GCM-simulated streamflows in the Manicouagan-5 watershed. The changes of hydrological variables were sampled through the Monte-Carlo approach based on the weights calculated using raw (RQ) or bias-corrected (DQ) GCM-simulated streamflows. (The depth of pink represents the level of inequality of the weights.)**