# Peer review of "Does the weighting of climate simulations result in a more reasonable quantification of hydrological impacts?"

_Hydrology and Earth System Sciences, 2019_

## Referee Comment (RC1) · Anonymous Referee #1 · 22 Feb 2019

This is a well written paper that studies the added value of weighting GCMs within an ensemble as a function of hydrological performance rather than as a function of climatological performance as usually done. The paper discusses some interesting aspects (e.g. the difference in outcome if weighting according to precipitation or temperature under different hydrological regimes) and comes to the conclusion that if raw GCM data is to be used, ensembles should be weighted based on streamflow rather than temperature or precipitation. In exchange, there is not much added value with streamflow-based weighting if the underlying GCMs are duly bias corrected. This outcome is not entirely surprising (see detailed comments) but I think it is nevertheless interesting for the readers of HESS and thus worth publishing.

Detailed comments In this paper, the GCM weighing is tested for large catchments (» 10'000 km2) that are simulated with a lumped model (GR4J) at a daily time step. With such a lumped model, it can a priori be assumed that the most important aspect of climate inputs for hydrological model performance and for future simulations is the actual precipitation and temperature bias. In fact, there is a whole body of hydrological literature on the importance of correct area-average precipitation estimates, which should perhaps be linked to this study. A starting point is the work of Lebel et al. 1987. Since the model is lumped, spatial differences between meteorological inputs derived from GCMs cannot show up in the simulation results otherwise than affecting the catchment-scale average values (i.e. the bias). Differences between GCM outputs in in temporal variability do most likely not show up because they are dampened by the model. The authors argue that the response of a catchment to climate input is nonlinear. This holds in general but if such a simple model is used, no surprising outcomes can be expected (not much difference between climate-based weighting and hydrological weighting in absence of major meteorological biases). This is a limitation of the study: major differences between climate-based weighting and hydrological weighting can a priori not be expected in the bias corrected set-up with such a simple model. This has to be discussed in sufficient detail in the paper and highlighted also in the perspectives. Finally: I am not an expert on bias correction methods. Accordingly I can only assume that this part of the work is state-of-the-art.

References Lebel, T., Bastin, G., Obled, C., and Creutin, J. D.: On the accuracy of areal rainfall estimation - a case study, Water Resources Research, 23, 2123-2134, 10.1029/WR023i011p02123, 1987.

---

## Referee Comment (RC2) · Anonymous Referee #2 · 20 Mar 2019

Summary and General Comments

The manuscript by Wang et al investigates the impact of multiple ensemble weighting techniques on the simulations of hydrological impacts for two different river basins. The authors compare the results from a hydrological impact model driven by weighted and unweighted GCM projections. In addition, the authors compare the results from bias-correcting the GCM output before weighting or not. They conclude that weighting the bias corrected GCM output has not a large effect while differences are larger when using raw output, improving the representation of the mean hydrograph and reducing the annual streamflow bias. The authors conclude that the equal weighting method is

a conservative approach and still viable given the small effect weighting has on a bias corrected ensemble.

Overall the paper is well and comprehensively written and the analysis extensive. The fact that weighting the bias corrected ensemble has a very small effect is not surprising. Given that bias correcting and weighting for performance have the same goal, bringing the ensemble closer to observed values, I do not understand why you would do both? Some of the risks and disadvantages for weighting, which are all true, also apply for bias correction (e.g. Sippel et al. 2016, Maraun et al. 2017). Both tools need to be applied carefully and have their pitfalls. For instance, it has been shown that out-of-sample testing is crucial for any kind of weighting or sub-sampling (e.g. Herger et al. 2018, Abramowitz et al. 2019), which is still missing so far in this study. In that sense I am not convinced that the authors come to the correct conclusion, even though their arguments are generally not wrong (see below). Weighting a GCM ensemble will conserve dependencies between different variables in a physically consistent way, and in cases where this is important, it might be preferred over bias correction. However, all the risks the authors mention apply, and the study shows nicely that there are still many open questions on how to use these methods properly. I would recommend to rephrase some of the discussion and conclusions more carefully and also account for the assumptions and risks in bias correction.

Specific comments

P1, L23-25: This conclusion is a bit far fetched and ignores the independence issue nicely described on page 2, around line 20. P4, L8: not relevant. P6, L20: The climatological mean of what? Temperature, precipitation, streamflow? All of them together or only individual? That makes a large difference and it has been shown that only using one at a time for PI is risky (Lorenz et al. 2018). P9, L19-23: Yes, but the same assumption applies for bias-correction. P10, L2-5: The testing is all done in sample. Out-of-sample testing is needed. P10, L9-11: At least for PI any metric could be considered, the fact that only climatology was used is because the authors chose

to do it this way, but is not a property of the method. P11-P12, L31-4: While these arguments are true, bias-correction has similar problems. Also, it looks to me that the equally weighted ensemble has the same issue? P12, L27-28: I do not think the results fully support this statement. We might not have found a clearly better way than model democracy, but equal weighting is as at least as arbitrary as weighting. P13, L1-2: Equally weighting is also arbitrary, given that is assumes all models are equally likely and independent, which they are not. P13, L5-10: Again, the same applies to bias-correction. P13, L11: Again, because you chose to only include one metric does not make it a property of the method. At least some of the weighting methods can account for multiple metrics to be included and people argue to do so (e.g. Knutti et al. 2017).

References: Abramowitz, G., et al: ESD Reviews: Model dependence in multi-model climate ensembles: weighting, sub-selection and out-of-sample testing, Earth Syst. Dynam., 10, 91-105, https://doi.org/10.5194/esd-10-91-2019, 2019.

Herger, N., et al: Selecting a climate model subset to optimise key ensemble properties, Earth Syst. Dynam., 9, 135–151, https://doi.org/10.5194/esd-9-135-2018, 2018.

Knutti, R., et al.: A climate model projection weighting scheme accounting for performance and interdependence, Geophys. Res. Lett., 44, 1909–1918, https://doi.org/10.1002/2016GL072012, 2017.

Lorenz, R., et al.: Prospects and caveats of weighting climate models for summer maximum temperature projections over North America, J. Geophys. Res.-Atmos., 123, 4509–4526, https://doi.org/10.1029/2017JD027992, 2018. Maraun D. et al: Towards process-informed bias correction of climate change simulations, Nature Climate Change, 7, 764–773, https://www.nature.com/articles/nclimate3418, 2017.

Sippel et al.: A novel bias correction methodology for climate impact Simulations, Earth Syst. Dynam., 7, doi:10.5194/esd-7-71-2016, 71–88, 2016.

24, 2019.

---

## Referee Comment (RC3) · Anonymous Referee #3 · 3 Apr 2019

This study applies different combinations of bias-correction (BC) and model weighting (MW) to post-process climate and hydrological projections in two catchments. Both BC and MW are receiving sustained attention in the community, and so far only few studies combine both. What is important to stress, is that although the underpinnings of these two approaches are quite different, their aim is arguably quite similar: close the gap between simulations and observations. This leads me to comment on the two main findings of the study:

Finding 1: "when using raw GCM outputs with no bias correction, streamflow-based weights better represent the mean hydrograph and reduce the bias of annual streamflow" P1L19-20: in my view, this is a natural consequence of applying MW, and in a way, it means that MW is used to correct for/mitigate climate model biases.

Finding 2: "when applying bias correction to GCM simulations before driving the hydrological model, the climate simulations become rather close to the observed climate, so that compared to equal weighting, the streamflow-based weights do not bring significant differences in the multi-model ensemble mean" P1L21-23: my interpretation is that employing successively two techniques with the same purpose makes the second technique redundant. Reducing the biases in the climate simulations, and then applying MW, makes it extremely difficult for the MW to discriminate between good and poor models. I recognise that BC is applied to the climate simulations and MW to the hydrological simulations, but since all the climate simulations are run through the same hydrological model, calibrated presumably with the forcing dataset also used to perform the BC, the differences in the streamflow simulations are minimal (as shown in Figure 3c and especially 4c). This lack of differences explains why the different weighting methods lead to similar results under current climate (the simulations are almost the same, so how they are combined makes little difference).

Overall, I suggest shifting the focus from current climatic conditions (for which no climate model and hence MW or BC is necessary) to future conditions (which rely on climate model simulations, which may need BC/MW). In my view, the focus is currently too much on the current conditions. For instance, in the abstract, the authors write "when applying bias correction to GCM simulations before driving the hydrological model, the climate simulations become rather close to the observed climate". This is true because of the nature of bias-correction, and was shown in previous studies (e.g., Hakala et al., 2018). What the grey area in Figures 3d and 4d tells us, however, is that under future conditions, there is substantial spread among the hydrological simulations, although the driving GCM simulations have been bias-corrected (likely because of the different sensitivities of the climate models).

Is there any way to apply MW based on these projected changes, and not based on

the streamflow simulations under current climate? In other words, are some of these projections more reliable than others and/or are some projections interdependent, and should be downweighted?

In summary, my impression is that Finding 1 is relevant but quite foreseeable. I think that Finding 2 is to a great extent due to the experimental design, in particular to the decision to apply BC and MW successively. I encourage the authors to rethink how to best combine MW and BC, for instance by using different periods and/or criteria for the MW.

Suggested reference

Hakala, K., Addor, N. and Seibert, J.: Hydrological modeling to evaluate climate model simulations and their bias correction, J. Hydrometeorol., 19, 1321–1337, doi:10.1175/JHM-D-17-0189.1, 2018.

---

## Author Comment (AC1) · 29 Apr 2019

Dear Anonymous Referee #1, We sincerely appreciate the referee's comments on this manuscript. These comments are all helpful to improve this manuscript. We have carefully studied and responded to all comments point-by-point. Please check the attached replies. The manuscript will be modified correspondingly.

Please also note the supplement to this comment:
https://www.hydrol-earth-syst-sci-discuss.net/hess-2019-24/hess-2019-24-AC1-supplement.pdf

[Figure]

**Supplement:**

Replies to Referee #1

**Does the weighting of climate simulations result in a more reasonable quantification of hydrological impacts?**

Hui-Min Wang, Jie Chen, Chong-Yu Xu, Hua Chen, Shenglian Guo, Ping Xie, Xiangquan Li

We sincerely appreciate the referee's comments on this manuscript. These comments are all helpful to improve this manuscript. We have carefully studied and responded to all comments point-by-point as follows. For clarity, all comments are given in *italics* and responses are given in plain text. The manuscript will be modified correspondingly.

*This is a well written paper that studies the added value of weighting GCMs within an ensemble as a function of hydrological performance rather than as a function of climatological performance as usually done. The paper discusses some interesting aspects (e.g. the difference in outcome if weighting according to precipitation or temperature under different hydrological regimes) and comes to the conclusion that if raw GCM data is to be used, ensembles should be weighted based on streamflow rather than temperature or precipitation. In exchange, there is not much added value with streamflow-based weighting if the underlying GCMs are duly bias corrected. This outcome is not entirely surprising (see detailed comments) but I think it is nevertheless interesting for the readers of HESS and thus worth publishing.*

We appreciate that the referee is in favor of the content of this research. Detailed comments have been replied as follows and will be addressed in the revision.

*Detailed comments*

*In this paper, the GCM weighing is tested for large catchments (» 10'000 km²) that are simulated with a lumped model (GR4J) at a daily time step. With such a lumped model, it can a priori be assumed that the most important aspect of climate inputs for hydrological model performance and for future simulations is the actual precipitation and temperature bias. In fact, there is a whole body of hydrological literature on the importance of correct area-average precipitation estimates, which should perhaps be linked to this study. A starting point is the work of Lebel et al. 1987. Since the model is lumped, spatial differences between meteorological inputs derived from GCMs cannot show up in the simulation results otherwise than affecting the catchment scale average values (i.e. the bias). Differences between GCM outputs in temporal variability do most likely not show up because they are dampened by the model. The authors argue that the response of a catchment to climate input is nonlinear. This holds in general but if such a simple model is used, no surprising outcomes can be expected (not much difference between climate-based weighting and hydrological weighting in absence*

*of major meteorological biases). This is a limitation of the study: major differences between climate-based weighting and hydrological weighting can a priori not be expected in the bias corrected set-up with such a simple model. This has to be discussed in sufficient detail in the paper and highlighted also in the perspectives. Finally: I am not an expert on bias correction methods. Accordingly I can only assume that this part of the work is state-of-the-art.*

Thanks for the comment. We agree with the referee that it is a limitation that only large watersheds and one lumped hydrological model were considered in this study. When using a lumped hydrological model, the nonlinear relationship between the climate variables and the impact variable (streamflow) may not be sufficiently revealed. Spatial differences between different climate simulations only affect the basin-averaged inputs to the hydrological model but not directly affect the process of runoff generation and streamflow routing. Temporal variations of climate simulations may be partially reduced by the lumped hydrological model. With the help of other more sophisticated hydrological models (such as the distributed model, SWAT), the differences between climate-based weights and streamflow-based weights may become more obvious. For the experiment of raw GCM-simulated streamflows, the weights based on streamflow show better performances than those based on climate variables. This may be related to large differences between climate simulations. But in the experiment of bias-corrected GCM-simulated streamflows, no much differences in the performances between unequal and equal weighting may be because only a simple hydrological model is used. In other words, even though the performance of the bias correction method differs in climate model simulations, the remaining biases among corrected climate simulations may not be well presented in streamflow simulations. All analyses above will be presented in the revision of the manuscript.

In addition, the daily bias correction (DBC) method used in this study has been applied in many recent studies (e.g., Chen et al., 2017; Li et al., 2019). It can be considered as a superior bias correction method in terms of correcting the bias of precipitation frequency and the bias in the distributions of precipitation amounts and temperature.

**References**

Chen, J., Brissette, F. P., Lucas-Picher, P., and Caya, D.: Impacts of weighting climate models for hydro-meteorological climate change studies, Journal of Hydrology, 549, 534-546, https://doi.org/10.1016/j.jhydrol.2017.04.025, 2017.

Li, L., Shen, M., Hou, Y., Xu, C.-Y., Lutz, A. F., Chen, J., Jain, S. K., Li, J., and Chen, H.: Twenty-first-century glacio-hydrological changes in the Himalayan headwater Beas River basin, Hydrology and Earth System Sciences, 23, 1483-1503, https://doi.org/10.5194/hess-23-1483-2019, 2019.

---

## Author Comment (AC2) · 29 Apr 2019

Replies to Referee #2

**Does the weighting of climate simulations result in a more reasonable quantification of hydrological impacts?**

Hui-Min Wang, Jie Chen, Chong-Yu Xu, Hua Chen, Shenglian Guo, Ping Xie, Xiangquan Li

We would like to thank the reviewer for the time taken in reviewing this paper. All comments are all valuable to improve this manuscript. Please find the point-by-point responses below. For clarity, comments are given in *italics*, and our responses are given in plain text. We will make the revisions to the manuscript as suggested.

> *Summary and General Comments*
>
> *The manuscript by Wang et al investigates the impact of multiple ensemble weighting techniques on the simulations of hydrological impacts for two different river basins. The authors compare the results from a hydrological impact model driven by weighted and unweighted GCM projections. In addition, the authors compare the results from bias-correcting the GCM output before weighting or not. They conclude that weighting the bias corrected GCM output has not a large effect while differences are larger when using raw output, improving the representation of the mean hydrograph and reducing the annual streamflow bias. The authors conclude that the equal weighting method is a conservative approach and still viable given the small effect weighting has on a bias corrected ensemble.*

We would like to express gratitude to the referee for reviewing this manuscript and offering precious comments and suggestions. All the comments and suggestions have been replied to below and will be addressed in the revision of manuscript.

> *Overall the paper is well and comprehensively written and the analysis extensive. The fact that weighting the bias corrected ensemble has a very small effect is not surprising. Given that bias correcting and weighting for performance have the same goal, bringing the ensemble closer to observed values, I do not understand why you would do both? Some of the risks and disadvantages for weighting, which are all true, also apply for bias correction (e.g. Sippel et al. 2016, Maraun et al. 2017). Both tools need to be applied carefully and have their pitfalls. For instance, it has been shown that out-of-sample testing is crucial for any kind of weighting or sub-sampling (e.g. Herger et al. 2018, Abramowitz et al. 2019), which is still missing so far in this study. In that sense I am not convinced that the authors come to the correct conclusion, even though their arguments are generally not wrong (see below). Weighting a GCM ensemble will conserve dependencies between different variables in a physically consistent way, and in cases where this is important, it might be preferred over bias correction. However, all the risks*

*the authors mention apply, and the study shows nicely that there are still many open questions on how to use these methods properly. I would recommend to rephrase some of the discussion and conclusions more carefully and also account for the assumptions and risks in bias correction.*

We agree with the referee that bias correction methods have similar goals as most of the weighting methods, which is to bring ensembles closer to observed values. Nevertheless, they still have different traits and functions. The bias correction directly deals with the biases of climate simulations and bridges the gap between the coarse outputs of climate models and data requirements of hydrological models. The model weighting assigns relative reliability to each climate simulation and aggregates multi-model ensembles. There are some differences between climate simulations whether the bias correction is done or not. In this case, a model weighting method always needs to be determined in order to obtain the overall impact evaluation and relevant uncertainty. Although the equal weighting is usually used in hydrological impact studies, it still deserves detailed investigation whether an unequal weighting method is necessary for bias-corrected ensembles. This problem is also mentioned in other studies (Alder and Hostetler, 2019; Chen et al., 2017). In addition, we agree with the referee that the bias correction has some similar risks to the model weighting. But the main goal of this study is to investigate the effects of model weighting when the bias correction is or is not conducted. We agree that these risks and problems of bias correction should be still addressed in the manuscript.

As suggested by the referee, we have added out-of-sample testing when evaluating performances of different weighting methods. The performances of weighting methods in this case are similar to the previous results that are based on historical observation. This confirms the conclusion of this study. Detailed results and relevant analysis will be presented in the revised manuscript.

*P1, L23-25: This conclusion is a bit far-fetched and ignores the independence issue nicely described on page 2, around line 20.*

Thanks for the referee's comment on this conclusion. We agree that this conclusion neglects the strengths of model weighting in impact studies and excessively trusts the function of bias correction. Actually, whether the bias correction overmatches the model weighting is not the research problem of this study, and we should focus on the effects of model weighting in two conditions (whether the bias correction is done or not). Thus, this conclusion will be fixed to "The equal weighting method may still be a viable and conservative choice when bias correction is conducted in the studies of hydrological climate change impacts".

Thanks for the comment. We agree that the introduction to the characteristics of the Daniel-Johnson Dam is redundant, but we also think that it is necessary to mention the Daniel-Johnson dam because the discharge data used for calibrating the hydrological model is collected here and is the inflow of the reservoir. This sentence will be shortened to "The outlet of the Manicouagan-5 River is the Daniel-Johnson Dam".

*P6, L20: The climatological mean of what? Temperature, precipitation, streamflow? All of them together or only individual? That makes a large difference and it has been shown that only using one at a time for PI is risky (Lorenz et al. 2018).*

Thanks for the comment on the presentation of methodology. We failed to state it clear enough. In this sentence, the climatological mean is for streamflow as indicated by P7, L1-3. Since GCM's performances on hydrological simulation are related to multiple variables (such as precipitation and temperatures in this study) and there is no widely accepted way to combine multiple sets of weights into single one, this study proposed to determine weights based on streamflow series. In this way, weighting based on streamflow simulations can synthesize GCMs' performances in both temperature and precipitation and circumvents the problem of non-linear relationship between climate and impact variables. In addition, calculating weights based on temperature and precipitation is also used in this study for comparison, as stated in P7, L1-3. Herein, the used variable will be stated in this sentence in the revised manuscript for clarity.

*P9, L19-23: Yes, but the same assumption applies for bias-correction.*

As stated in P9, L15-17, we agree with the referee that the assumption of stationary biases in GCM outputs also applies for the bias correction method. In this sentence, we intend to state that if the weighting methods still follow the same assumption as the bias correction (as most performance-based weighting methods do), there will be no needs to do unequal weighting. However, some other weighting methods contain other criteria that do not follow the same assumption, such as the interdependence criterion in the PI method and the future convergence criterion in the REA method. This point will be rephrased in the revised manuscript.

*P10, L2-5: The testing is all done in sample. Out-of-sample testing is needed.*

Thanks for the suggestion. We agree with the reviewer and we have done the out-of-sample testing by conducting model-as-truth experiments (Herger et al., 2018). Relevant results and analyses will be added and discussed in the revised version of the manuscript.

*P10, L9-11: At least for PI any metric could be considered, the fact that only climatology was used is because the authors chose to do it this way, but is not a property of the method.*

We appreciate the referee's comments on this analysis. We approve of the idea that other metrics can be used in PI method. In fact, different metrics can also be applied into some other weighting

methods if users want to do so, even though these methods are designed to use the climatological mean. However, many researches and end-users in hydrological impacts only consider the climatological mean (e.g., Wilby and Harris, 2006; Chen et al., 2017). In this sentence, we were to express that the different performances on different metrics may be due to the usage of weighting methods by end-users, which we failed to state clearly in the submitted version of the manuscript. Thus, this sentence will be corrected to "This may be because only the mean value (climatological mean or monthly mean series) was used as the evaluation metric when determining weights in this study, and peak or extreme values were not considered".

In addition, using different metrics may result in different performances of a weighting method (as stated in P12, L29-31). However, the main focus of this study is the effects of weighting GCM based on their performances in streamflow simulations. Whether other metrics bring about different results needs further research and is beyond the scope of this study. Therefore, in the Discussion section, we will add some discussion on the metric adopted for the weighting methods.

> *P11-P12, L31-4: While these arguments are true, bias-correction has similar problems. Also, it looks to me that the equally weighted ensemble has the same issue?*

We agree with the referee that similar to the model weighting, the bias-correction has the problem of non-linear relationship between climate variables and impact variables. However, the bias-correction does not have the problem of trade-off among different climate variables. This is because bias correction is done for each variable, and corrected variables are then inputted to the impact model at the same time. No trade-off needs to be processed in this procedure. For model weighting methods, how to combine different sets of climate-based weights becomes a question. For example, the weights calculated based on temperature and precipitation need to be combined into a single set when generalizing the hydrological impacts for the two watersheds in this study. In this case, the trade-off between two variables is needed, which may be varied in different watersheds.

Similarly, the equal weighting is the same. The trade-off between different variables is not needed, but it also cannot circumvent the problem of non-linear relationship between climate and impact variables. Therefore, as stated in P13, L18, the equal weighting is only a conservative option for handling multi-model ensembles in impact studies. All these problems will be discussed in the revised manuscript.

> *P12, L27-28: I do not think the results fully support this statement. We might not have found a clearly better way than model democracy, but equal weighting is as at least as arbitrary as weighting.*

We thank the referee for this comment. We agree that this statement is somewhat ambitious for the results. This statement will be fixed to "When GCM outputs are processed by the bias correction, compared to the equal weighing method, unequal weighting methods do not bring about much

different impact results".

*P13, L1-2: Equally weighting is also arbitrary, given that is assumes all models are equally likely and independent, which they are not.*

We appreciate the referee's comment. As stated in P2, L14-23, equal weighting ignores the differences in the performances and potential dependency of GCMs. But at the same time, unequal weighting methods also have potential problems of reducing projection accuracy and concealing projection uncertainty (as stated in P13, L2-4). Therefore, equal weighting should not be regarded as the final solution but a conservative method, and the weighting methods should be used with cautions for now. Accordingly, this sentence will be corrected as: "All of these aspects in weighting methods are often predefined without detailed examination or based on expert experience and, thus, can actually introduce several layers of subjective uncertainty. Notwithstanding the equal weighting is not a totally perfect solution, model weighting methods should be used with cautions and the results of equal weighting should be presented along with the results of unequal weighting methods".

We agree that the bias correction has the same problem that climate simulations are corrected statically. Herein, we did not intend to say that bias correction methods have some strengths over the model weighting but only to state one problem of present model weighting methods. This could be a focus for future study of model weighting. In order to stress this problem and eliminate vagueness, the statement here will be modified as follows.

Weights are generally assigned to climate simulations in a static way (i.e. weights in the reference period are the same as those in the future period). This usage shares the same assumption with bias-correction methods that the performances of GCM simulations are stable and stationary. However, some studies have shown that model skills are nonstationary in a changing climate, and models with better performance in the reference period do not necessarily provide more realistic signals of climate change. The way to deal with the dynamic reliability of climate models deserves further study.

*P13, L11: Again, because you chose to only include one metric does not make it a property of the method. At least some of the weighting methods can account for multiple metrics to be included and people argue to do so (e.g. Knutti et al. 2017).*

We agree with the referee that in the PI method, multiple metrics could be used to weight climate simulations. Yet, when introducing multiple metrics, there must be decisions on the relevant diagnostic metrics and the way to synthesize GCM's overall performances in multiple metrics. Some studies have stated that using calibrated multiple metrics helps to improve the agreement with observation (Knutti et al., 2017; Lorenz et al., 2018), while some argue that multiple metrics form

another level of uncertainty within weighting methods (Christensen et al., 2010). These problems deserve further detailed investigation but they are beyond the scope of this study (which is to investigate whether weighting based on streamflow simulations induces better quantification of hydrological impacts). Thus, this sentence will be modified correspondingly to mention the use of multiple metrics and its potentials to strengthen weighted results.

**References**

Alder, J. R., and Hostetler, S. W.: The Dependence of Hydroclimate Projections in Snow‑Dominated Regions of the Western United States on the Choice of Statistically Downscaled Climate Data, Water Resources Research, 55, 2279-2300, https://doi.org/10.1029/2018wr023458, 2019.

Chen, J., Brissette, F. P., Lucas-Picher, P., and Caya, D.: Impacts of weighting climate models for hydro-meteorological climate change studies, Journal of Hydrology, 549, 534-546, https://doi.org/10.1016/j.jhydrol.2017.04.025, 2017.

Christensen, J. H., Kjellström, E., Giorgi, F., Lenderink, G., and Rummukainen, M.: Weight assignment in regional climate models, Climate Research, 44, 179-194, https://doi.org/10.3354/cr00916, 2010.

Herger, N., Abramowitz, G., Knutti, R., Angélil, O., Lehmann, K., and Sanderson, B. M.: Selecting a climate model subset to optimise key ensemble properties, Earth System Dynamics, 9, 135-151, https://doi.org/10.5194/esd-9-135-2018, 2018.

Knutti, R., Sedláček, J., Sanderson, B. M., Lorenz, R., Fischer, E. M., and Eyring, V.: A climate model projection weighting scheme accounting for performance and interdependence, Geophysical Research Letters, https://doi.org/10.1002/2016gl072012, 2017.

Lorenz, R., Herger, N., Sedláček, J., Eyring, V., Fischer, E. M., and Knutti, R.: Prospects and Caveats of Weighting Climate Models for Summer Maximum Temperature Projections Over North America, Journal of Geophysical Research: Atmospheres, 123, 4509-4526, https://doi.org/10.1029/2017jd027992, 2018.

Wilby, R. L., and Harris, I.: A framework for assessing uncertainties in climate change impacts: Low-flow scenarios for the River Thames, UK, Water Resources Research, 42, W02419, https://doi.org/10.1029/2005wr004065, 2006.

---

## Author Comment (AC3) · 29 Apr 2019

Dear Anonymous Referee #3, We sincerely appreciate the referee's comments and suggestions on the manuscript. All suggestions are helpful to improve this manuscript. We have carefully studied, considered and responded to all comments point-by-point. Please check the attached replies. The manuscript will be modified accordingly.

Please also note the supplement to this comment:
https://www.hydrol-earth-syst-sci-discuss.net/hess-2019-24/hess-2019-24-AC3-supplement.pdf

[Figure]

**Supplement:**

Replies to Referee #3

**Does the weighting of climate simulations result in a more reasonable quantification of hydrological impacts?**

Hui-Min Wang, Jie Chen, Chong-Yu Xu, Hua Chen, Shenglian Guo, Ping Xie, Xiangquan Li

We sincerely appreciate the referee's comments and suggestions on the manuscript. All suggestions are helpful to improve this manuscript. We have carefully studied, considered and responded to all comments point-by-point. For clarity, all comments are given in *italics* and responses are given in plain text. The manuscript will be modified accordingly.

> *This study applies different combinations of bias-correction (BC) and model weighting (MW) to post-process climate and hydrological projections in two catchments. Both BC and MW are receiving sustained attention in the community, and so far only few studies combine both. What is important to stress, is that although the underpinnings of these two approaches are quite different, their aim is arguably quite similar: close the gap between simulations and observations. This leads me to comment on the two main findings of the study:*

We would like to thank the referee for the time taken in reviewing this manuscript. All comments have been replied to below and will be addressed in the revision.

> *Finding 1: "when using raw GCM outputs with no bias correction, streamflow-based weights better represent the mean hydrograph and reduce the bias of annual streamflow" P1L19-20: in my view, this is a natural consequence of applying MW, and in a way, it means that MW is used to correct for/mitigate climate model biases.*

Thanks for the comment. We agree with the referee that MW is used to mitigate biases, but this is not the specific focus of this study and we failed to state the conclusion clear enough. Actually, in this sentence, we intended to emphasize the advantages of streamflow-based weights over the weights calculated using climate variables (i.e. temperature and precipitation in this study). As stated in P12, L9-14, when dealing with the raw GCM-simulated streamflows, biases in multi-model mean of annual streamflow are reduced more by the weights based on the impact variable (streamflow), comparing with the weights based on climate variables. Herein, we will modify the expression of Finding 1 to make this point clearer.

> *Finding 2: "when applying bias correction to GCM simulations before driving the hydrological model, the climate simulations become rather close to the observed climate, so that compared to equal weighting, the streamflow-based weights do not bring significant differences in the multi-model ensemble mean" P1L21-23: my interpretation is that employing*

*successively two techniques with the same purpose makes the second technique redundant. Reducing the biases in the climate simulations, and then applying MW, makes it extremely difficult for the MW to discriminate between good and poor models. I recognise that BC is applied to the climate simulations and MW to the hydrological simulations, but since all the climate simulations are run through the same hydrological model, calibrated presumably with the forcing dataset also used to perform the BC, the differences in the streamflow simulations are minimal (as shown in Figure 3c and especially 4c). This lack of differences explains why the different weighting methods lead to similar results under current climate (the simulations are almost the same, so how they are combined makes little difference).*

We agree with the referee that in this study, MW loses the ability to discriminate the performances of climate simulations after the bias correction. This is also a finding of this study, which was mentioned in P12, L18-20. We will modify this sentence to make this point clearer.

In fact, MW is not designed for dealing with hydrological simulations but a necessary process to handle the ensemble of multiple climate simulations. Even after bias correction, there still exist some differences between climate simulations. In order to obtain evaluation of climate change impacts, it is unavoidable to choose a MW method to synthesize the simulation results from the ensemble (whether or not bias correction is done). Thus, MW is an indispensable process. Actually, both BC and MW are common procedures in regional impact studies. Although it is common to use equal weighting for bias-corrected ensembles, whether unequal weighting is the best choice remains to be investigated (Alder and Hostetler, 2019). The results of this study show that when the bias correction is done in impact studies, unequal weighting does not bring much difference to the impact evaluation. This supports the usage of equal weighting for bias-corrected ensembles so far. Nonetheless, we still think that with further development of weighting methods (e.g., more aggressive or multi-objective weighting methods), unequal weighting maybe helps to bring different or more reasonable consequences. The discussion on the weighting methods for the bias-corrected ensembles will be added in the revised manuscript.

*Overall, I suggest shifting the focus from current climatic conditions (for which no climate model and hence MW or BC is necessary) to future conditions (which rely on climate model simulations, which may need BC/MW). In my view, the focus is currently too much on the current conditions. For instance, in the abstract, the authors write "when applying bias correction to GCM simulations before driving the hydrological model, the climate simulations become rather close to the observed climate". This is true because of the nature of bias-correction, and was shown in previous studies (e.g., Hakala et al., 2018). What the grey area in Figures 3d and 4d tells us, however, is that under future conditions, there is substantial spread among the hydrological simulations, although the driving GCM simulations have been bias-corrected (likely because of the different sensitivities of the climate models).*

Thanks for the comment. We agree with the referee that more attention should be paid to the future projections. In this version of manuscript, future simulations are only evaluated in the form of uncertainty (Section 4.4), since there is no observation in the future period to be compared with. In order to partly overcome this problem, we have added the out-of-sample testing for this study following the suggestion of referee #2. In out-of-sample testing, the output of one climate model was regarded as the "truth" and the outputs of the remaining 28 climate models were used as simulations to this "truth" model. Then the weights were re-calculated for the remaining models. Since there is a "truth" result for the future period in this case, the performances of weighting methods in reproducing the future "truth" can be evaluated. In the out-of-sample testing, each climate model was regarded as truth in turn. In general, the results of out-of-sample testing are similar to the results using historical observations, which supports the conclusion of this study. The detailed results and analyses of out-of-sample testing will be added and discussed in the revised manuscript.

In addition, it is true that the differences between ensemble members have been greatly reduced during the reference period while there are still considerable differences in the future period (which had been stated in P9, L15-19). This may be because the bias of climate models is nonstationary (Hui et al., 2018). However, the sentence in the abstract is only an explanation to the results of Finding 2 instead of a focus of this study. But we failed to state this logic clear enough. Therefore, this sentence will be modified to make the focus of this study clearer.

> *Is there any way to apply MW based on these projected changes, and not based on the streamflow simulations under current climate? In other words, are some of these projections more reliable than others and/or are some projections interdependent, and should be downweighted?*

We thank the referee for this suggestion. Actually, the REA method in this study concludes projected values when assigning weights. The REA considers both smaller differences to the observation in the reference period and more concentrated projections in the future period. Although the weights calculated by the REA method are most differentiated for the bias-corrected ensemble (as Fig. 2 shows), they still bring little impacts on the final results of multi-model means. In addition, the PI method considers independency between climate simulations when determining weights, but it only relies on reference values which have been tuned by the bias-correction methods. The ability of independent criterion may fail because of the bias correction. Therefore, in the case of bias-corrected ensembles, some modifications may be needed for these MW methods to include future values. This point will be further discussed in the revised manuscript.

> *In summary, my impression is that Finding 1 is relevant but quite foreseeable. I think that Finding 2 is to a great extent due to the experimental design, in particular to the decision to apply BC and MW successively. I encourage the authors to rethink how to best combine MW*

*and BC, for instance by using different periods and/or criteria for the MW.*

We appreciate the comments from the referee. As presented in the last response, the out-of-sample testing will be added in the discussion as a complement. In addition, we will better state that the main focus of this study is to investigate the influences of MW methods on the evaluation of climate change impacts (when the bias correction is or is not done), and to study whether the weighting determined based on the impact variable (streamflow) can induce more reasonable results. This investigation is necessary because MW is a procedure to generalize the results of ensembles and the best way to do it remains questionable. This explanation to the usage of MW and BC will be added in the Discussion section.

**References**

Alder, J. R., and Hostetler, S. W.: The Dependence of Hydroclimate Projections in Snow‐Dominated Regions of the Western United States on the Choice of Statistically Downscaled Climate Data, Water Resources Research, 55, 2279-2300, https://doi.org/10.1029/2018wr023458, 2019.

Hui, Y., Chen, J., Xu, C. Y., Xiong, L., and Chen, H.: Bias nonstationarity of global climate model outputs: The role of internal climate variability and climate model sensitivity, International Journal of Climatology, 39, 2278-2294, https://doi.org/10.1002/joc.5950, 2018.

---

## Author Response (AR1)

Authors' responses to comments

**Does the weighting of climate simulations result in a more reasonable quantification of hydrological impacts?**

Hui-Min Wang, Jie Chen, Chong-Yu Xu, Hua Chen, Shenglian Guo, Ping Xie, Xiangquan Li

We would like to appreciate the editor's and the three anonymous referees' valuable suggestions and comments on the manuscript. These suggestions are helpful to improve this manuscript. We have carefully studied and responded to all comments point-by-point as follows. For clarity, all comments are given in *italics* and responses are given in plain text. The manuscript has been modified accordingly.

**Responses to Editor's comments**

We would like to thank the editor for reviewing this manuscript. Our responses are as follows.

> *Thank you for posting your responses to the three referees' reports. The reviews raised some important comments and suggestions that I urge you to consider as I believe they will improve the quality of the manuscript. Based on my own reading, I find this to be an interesting paper that could fit the scope of HESS and would be of interest to the community. I invite you to upload a revised manuscript, incorporating the proposed changes and additions, and making any other modifications where you see fit ('major revision').*

We appreciate that the editor is in favor of the content of this research. We have thoroughly studied the comments from all referees, and the manuscript has been revised accordingly. Specific responses to referees' comments have been listed below. Revised manuscript and specific changes to the manuscript have also been attached at the end of the present response.

> *In addition to the comments from the reviewers, I kindly ask you to add "Author contribution", "Data availability" and "Competing interests" sections to the manuscript, as indicated in the guidance for authors.*

Thanks for the suggestion. We have added relevant information at the end of revised manuscript followed the guidance for authors [P15, L21-P16, L2].

**Responses to Referee #1's comments**

We sincerely appreciate the referee's comments and suggestions on the manuscript. Our responses

are as follows.

*This is a well written paper that studies the added value of weighting GCMs within an ensemble as a function of hydrological performance rather than as a function of climatological performance as usually done. The paper discusses some interesting aspects (e.g. the difference in outcome if weighting according to precipitation or temperature under different hydrological regimes) and comes to the conclusion that if raw GCM data is to be used, ensembles should be weighted based on streamflow rather than temperature or precipitation. In exchange, there is not much added value with streamflow-based weighting if the underlying GCMs are duly bias corrected. This outcome is not entirely surprising (see detailed comments) but I think it is nevertheless interesting for the readers of HESS and thus worth publishing.*

We appreciate that the referee is in favor of the content of this research. Detailed comments have been replied as follows and addressed in the revision.

*Detailed comments*

*In this paper, the GCM weighing is tested for large catchments (» 10'000 km$^2$) that are simulated with a lumped model (GR4J) at a daily time step. With such a lumped model, it can a priori be assumed that the most important aspect of climate inputs for hydrological model performance and for future simulations is the actual precipitation and temperature bias. In fact, there is a whole body of hydrological literature on the importance of correct area-average precipitation estimates, which should perhaps be linked to this study. A starting point is the work of Lebel et al. 1987. Since the model is lumped, spatial differences between meteorological inputs derived from GCMs cannot show up in the simulation results otherwise than affecting the catchment scale average values (i.e. the bias). Differences between GCM outputs in temporal variability do most likely not show up because they are dampened by the model. The authors argue that the response of a catchment to climate input is nonlinear. This holds in general but if such a simple model is used, no surprising outcomes can be expected (not much difference between climate-based weighting and hydrological weighting in absence of major meteorological biases). This is a limitation of the study: major differences between climate-based weighting and hydrological weighting can a priori not be expected in the bias corrected set-up with such a simple model. This has to be discussed in sufficient detail in the paper and highlighted also in the perspectives. Finally: I am not an expert on bias correction methods. Accordingly I can only assume that this part of the work is state-of-the-art.*

Thanks for the comment. We agree with the reviewer that when using a lumped model, the nonlinear relationship between the climate variables and the impact variable (streamflow) may not be sufficiently revealed. Spatial differences between different climate simulations only affect the basin-averaged inputs to the hydrological model but not directly affect the process of runoff generation

and streamflow routing (Lebel et al., 1987). Temporal variations of climate simulations may be partially reduced by the lumped hydrological model as well. With the help of other more sophisticated hydrological models (such as distributed models), the differences between climate-based weights and streamflow-based weights may become more obvious. For the experiment of raw GCM-simulated streamflows, the weights based on streamflow show better performances than those based on climate variables. This may be related to large differences among climate simulations. But in the experiment of streamflows simulated using bias-corrected GCM outputs, no much discrepancy in the performances between unequal and equal weighting may be partly because only a simple hydrological model is used. In other words, the remaining differences among corrected climate simulations may not be well presented in streamflow simulations when a lumped hydrological model is used in such large watersheds. All the analyses above have been presented in the Discussion section of revised manuscript [P14, L23-P15, L2].

In addition, the daily bias correction (DBC) method used in this study has been applied in many recent studies (e.g., Chen et al., 2017; Li et al., 2019). It can be considered as a superior bias correction method in terms of correcting the bias of precipitation frequency and the bias in the distributions of precipitation amounts and temperature.

**Responses to Referee #2's comments**

We would like to thank the referee for the time taken in reviewing our paper. Please find the point-by-point responses below. We have made revisions to the manuscript as suggested.

> *Summary and General Comments*
>
> *The manuscript by Wang et al investigates the impact of multiple ensemble weighting techniques on the simulations of hydrological impacts for two different river basins. The authors compare the results from a hydrological impact model driven by weighted and unweighted GCM projections. In addition, the authors compare the results from bias-correcting the GCM output before weighting or not. They conclude that weighting the bias corrected GCM output has not a large effect while differences are larger when using raw output, improving the representation of the mean hydrograph and reducing the annual streamflow bias. The authors conclude that the equal weighting method is a conservative approach and still viable given the small effect weighting has on a bias corrected ensemble.*

We would like to express gratitude to the referee for reviewing this manuscript and for the professional summary of the work. All the comments and suggestions have been replied to below and addressed in the revision of manuscript.

*Overall the paper is well and comprehensively written and the analysis extensive. The fact that weighting the bias corrected ensemble has a very small effect is not surprising. Given that bias correcting and weighting for performance have the same goal, bringing the ensemble closer to observed values, I do not understand why you would do both? Some of the risks and disadvantages for weighting, which are all true, also apply for bias correction (e.g. Sippel et al. 2016, Maraun et al. 2017). Both tools need to be applied carefully and have their pitfalls. For instance, it has been shown that out-of-sample testing is crucial for any kind of weighting or sub-sampling (e.g. Herger et al. 2018, Abramowitz et al. 2019), which is still missing so far in this study. In that sense I am not convinced that the authors come to the correct conclusion, even though their arguments are generally not wrong (see below). Weighting a GCM ensemble will conserve dependencies between different variables in a physically consistent way, and in cases where this is important, it might be preferred over bias correction. However, all the risks the authors mention apply, and the study shows nicely that there are still many open questions on how to use these methods properly. I would recommend to rephrase some of the discussion and conclusions more carefully and also account for the assumptions and risks in bias correction.*

We agree with the referee that bias correction methods have similar goals as most of the weighting methods, which are to bring ensembles closer to observed values. Nevertheless, they still have different traits and functions. The bias correction directly deals with the biases of climate simulations and bridges the gap between the coarse outputs of climate models and data requirements of hydrological models. The model weighting assigns relative reliability to each climate simulation and aggregates multi-model ensembles. There are some differences between climate simulations whether the bias correction is done or not. In this case, a model weighting method always needs to be determined in order to obtain the overall impact evaluation and relevant uncertainty. Although the equal weighting is usually used in hydrological impact studies, it still deserves detailed investigation whether an unequal weighting method is necessary for bias-corrected ensembles. This problem is also mentioned in other studies (Alder and Hostetler, 2019; Chen et al., 2017). This point has been underscored in the revised manuscript [P2, L10-13 & P13, L12-14].

In addition, we agree with the referee that the bias correction has some similar risks to the model weighting. But the main goal of this study is to investigate the effects of model weighting when the bias correction is or is not conducted. To be sure, we agree that these risks and problems of bias correction should still be addressed in the manuscript. Relevant corrections have been made in the revised manuscript [P9, L18-20 & P14, L7-8].

As suggested by the referee, we have also added out-of-sample testing when evaluating performances of different weighting methods. The performances of weighting methods in this case are similar to the previous results that are based on historical observation. This confirms the conclusion of this study. Detailed results and relevant analysis have been presented in the detailed

responses to specific comments below.

*P1, L23-25: This conclusion is a bit far-fetched and ignores the independence issue nicely described on page 2, around line 20.*

Thanks for the referee's comment on this conclusion. We agree that this conclusion neglects the strengths of model weighting in impact studies and excessively trusts the function of bias correction. Actually, whether the bias correction overmatches the model weighting is not the research problem of this study, and we should focus on the effects of model weighting in two conditions (whether the bias correction is done or not). Thus, this conclusion has been fixed to "Thus, the equal weighting method may still be a viable and conservative choice when bias correction to GCM simulations is conducted in hydrological climate change impact studies" [P1, L23-25].

*P4, L8: not relevant.*

Thanks for the comment. We agree that the introduction to the characteristics of the Daniel-Johnson Dam is redundant, but we also think that it is necessary to mention the Daniel-Johnson dam because the discharge data used for calibrating the hydrological model is collected here and is the inflow of the reservoir. This sentence has been shortened to "The outlet of the Manicouagan-5 River is the Daniel-Johnson Dam" [P4, L9-10], and the data collection of the observed streamflow of Manicouagan-5 has been stated more clearly [P4, L20-21].

*P6, L20: The climatological mean of what? Temperature, precipitation, streamflow? All of them together or only individual? That makes a large difference and it has been shown that only using one at a time for PI is risky (Lorenz et al. 2018).*

Thanks for the comment on the presentation of methodology. We failed to state it clear enough. In this sentence, the climatological mean is for streamflow. Since GCM's performances on hydrological simulation are related to multiple variables (such as precipitation and temperatures in this study) and there is no widely accepted way to combine multiple sets of weights into single one, this study proposed to determine weights directly based on streamflow series. In this way, weighting based on streamflow simulations can synthesize GCMs' performances in both temperature and precipitation and circumvent the problem of non-linear relationship between climate and impact variables. In addition, calculating weights based on temperature and precipitation is also used in this study for comparison, as stated in P7, L2-3. Herein, the used variable has been stated more clearly [P6, L20-21].

*P9, L19-23: Yes, but the same assumption applies for bias-correction.*

As stated in P9, L18-20, we agree with the referee that the assumption of stationary biases in GCM

outputs also applies for the bias correction method. In this sentence, we intend to state that most weighting methods still follow the same assumption as the bias correction and do not overcome the potential problem (especially for the performance-based weighting methods). However, some other weighting methods contain other criteria that do not follow the same assumption, such as the interdependence criterion in the PI method and the future convergence criterion in the REA method. This point has been rephrased in the revised manuscript [P9, L18-23].

*P10, L2-5: The testing is all done in sample. Out-of-sample testing is needed.*

Thanks for the suggestion. We agree with the reviewer and we have done the out-of-sample testing by conducting model-as-truth experiments (Herger et al., 2018). In model-as-truth experiments, the output of one climate model was regarded as the "truth" and the outputs of the remaining 28 climate models were used as simulations to this "truth" model. Then the weights were re-calculated for these remaining models. Since there is a "truth" result for the future period in this case, the performances of weighting methods can be evaluated in terms of reproducing the future "truth". Note that each climate model was regarded as truth in turn, and the results are gathered for each climate simulation playing as the truth.

Figure R1 shows the results of out-of-sample testing over the Xiangjiang watershed for biases of weighted multi-model mean hydrological indices, which are the same as those in Fig. 5. The left and right sides of each stick respectively represent the biases at the reference and future periods when one climate model is regarded as the truth. Similar to Fig. 5, the bias of weighted mean being closer to 0 means that the corresponding weighting method performs better. In general, the results of out-of-sample testing are similar to those of using historical observations. For the experiment of streamflows simulated by raw GCM outputs, Fig. R1a-c shows that unequally weighted means more or less become closer to the truth simulation than those of equal weighting in both reference and future periods. The unequal streamflow-based weights can help to reduce the biases. In particular, the three methods with the most differentiated weights (REA, UREA and CPI) reduce more biases of annual streamflow when compared with other methods (i.e., the ranges of the biases calculated by these three methods are narrower and closer to 0 when different simulations are used as the truth). In addition, although the biases in the future period tend to be larger than those in the reference period, the weighted means still have a slight improvement in most cases. However, for the experiment of using bias-corrected GCM outputs to simulate streamflows, as shown by the similar patterns among equal and unequal weighting methods (Fig. R1d-f), the unequally weighted multi-model means have similar biases to those of equal weighting in both reference and future periods. In addition, the results of out-of-sample testing over the Manicouagan-5 watershed are shown in Fig. R2, and generally, they are also similar to the results of using observations (Fig. 6).

All results and analyses above have been added as a separate sub-section in the section of Results of the revised manuscript [P11, L25-P12, L17 & Fig. 9-10].

*P10, L9-11: At least for PI any metric could be considered, the fact that only climatology was used is because the authors chose to do it this way, but is not a property of the method.*

We appreciate the referee's comments on this analysis. We approve of the idea that other metrics can be used in PI method. In fact, different metrics can also be applied to some other weighting methods, even though these methods are designed to use the climatological mean. However, many researches and end-users in hydrological impacts only consider the climatological mean (e.g., Wilby and Harris, 2006; Chen et al., 2017). In this sentence, we were to express that the various performances of different metrics may be due to the usage of weighting methods by end-users, which we failed to state clearly in the original version of the manuscript. Thus, this sentence has been corrected [P10, L9-11] and this point has been discussed in the Discussion section [P14, L12-18].

In addition, using different metrics may result in different performances of a weighting method. However, the main focus of this study is on effects of weighting GCM based on their performances in streamflow simulations. Whether other metrics bring about different results needs further research and is beyond the scope of this study. Therefore, the discussion on the metric adopted for the weighting methods has been added in the revised version of manuscript [P14, L18-22].

*P11-P12, L31-4: While these arguments are true, bias-correction has similar problems. Also, it looks to me that the equally weighted ensemble has the same issue?*

We agree with the referee that similar to the model weighting, the bias-correction has the problem of non-linear relationship between climate variables and impact variables. However, the bias-correction does not have the problem of trade-off among different climate variables. This is because bias correction is done for each variable, and corrected variables are then simultaneously inputted to the impact model. No trade-off needs to be processed in this procedure. For model weighting methods, how to combine different sets of climate-based weights becomes a question. For example, the weights calculated based on temperature and precipitation need to be combined into a single set when generalizing the hydrological impacts for the two watersheds in this study. In this case, the trade-off between two variables is needed, which may be varied in different watersheds.

Similarly, the equal weighting is the same. The trade-off between different variables is not needed, but it also cannot circumvent the problem of non-linear relationship between climate and impact variables. Therefore, as stated in P1, L23-25, the equal weighting is only a conservative option for handling multi-model ensembles in hydrological impact studies.

*P12, L27-28: I do not think the results fully support this statement. We might not have found a clearly better way than model democracy, but equal weighting is as at least as arbitrary as weighting.*

We thank the referee for this comment. We agree that this statement is somewhat ambitious for the results. This statement has been fixed to "In this experiment, compared to the equal weighing method, unequal weighting methods do not bring about much disparateness to the results of hydrological impacts. It is still viable to attend to the bias-corrected ensembles with the equal weighting method" [P13, L24-27].

> *P13, L1-2: Equally weighting is also arbitrary, given that is assumes all models are equally likely and independent, which they are not.*

We appreciate the referee's comment. As stated in P2, L14-23, equal weighting ignores the differences in the performances and potential dependency of GCMs. But at the same time, unequal weighting methods also have potential problems of reducing projection accuracy and concealing projection uncertainty (as stated in P13, L34-P14, L3). Therefore, equal weighting should not be regarded as the final solution but a conservative method, and the weighting methods should be used with cautions for now. Accordingly, this sentence has been corrected as: "All of these aspects in weighting methods are often predefined without detailed examination or based on expert experience and, thus, can actually introduce several layers of subjective uncertainty. Notwithstanding the equal weighting is not a perfect solution, model weighting methods should be used with cautions and the results of equal weighting should be presented along with the results of unequal weighting methods" [P13, L33-P14, L5].

> *P13, L5-10: Again, the same applies to bias-correction.*

We agree that the bias correction has the same problem that climate simulations are corrected statically. Herein, we did not intend to say that bias correction methods are superior to model weighting methods but only to state one problem of present model weighting methods. This could be a focus for future study of model weighting. In order to stress this problem and eliminate vagueness, the statement here has been modified as follows [P14, L6-12].

Weights are generally assigned to climate simulations in a static way with an assumption that weights in the future period are the same as those in the reference period. This usage shares the same assumption with bias-correction methods that the performances of GCM simulations are stable and stationary. However, some studies have shown that model skills are nonstationary in a changing climate, and models with better performance in the reference period do not necessarily provide more realistic signals of climate change. The way to deal with the dynamic reliability of climate models deserves further study.

> *P13, L11: Again, because you chose to only include one metric does not make it a property of the method. At least some of the weighting methods can account for multiple metrics to be included and people argue to do so (e.g. Knutti et al. 2017).*

We agree with the referee that in the PI method, multiple metrics could be used to weight climate simulations. Yet, when introducing multiple metrics, there must be decisions on the relevant diagnostic metrics and the way to synthesize GCM's overall performances in multiple metrics. Some studies suggested using calibrated multiple metrics because it can improve the rationality of weighted multi-model mean (Knutti et al., 2017; Lorenz et al., 2018), while some argued that multiple metrics form another level of uncertainty within weighting methods (Christensen et al., 2010). These problems deserve further detailed investigation but they are beyond the scope of this study (which is to investigate whether weighting based on streamflow simulations induces better quantification of hydrological impacts). Thus, this sentence has been modified correspondingly to mention the use of multiple metrics and its potentials to strengthen weighted results [P14, L18-22].

**Responses to Referee #3's comments**

We would like to express gratitude to the referee for reviewing this paper and offering valuable suggestions. Please find the point-by-point responses below.

*This study applies different combinations of bias-correction (BC) and model weighting (MW) to post-process climate and hydrological projections in two catchments. Both BC and MW are receiving sustained attention in the community, and so far only few studies combine both. What is important to stress, is that although the underpinnings of these two approaches are quite different, their aim is arguably quite similar: close the gap between simulations and observations. This leads me to comment on the two main findings of the study:*

We would like to thank the referee for the time taken in reviewing our manuscript and for the professional summary of the work. All comments have been replied to below and has been addressed in the revision.

*Finding 1: "when using raw GCM outputs with no bias correction, streamflow-based weights better represent the mean hydrograph and reduce the bias of annual streamflow" P1L19-20: in my view, this is a natural consequence of applying MW, and in a way, it means that MW is used to correct for/mitigate climate model biases.*

Thanks for the comment. We agree with the referee that MW is used to mitigate biases, but this is not the specific focus of this study and we failed to state the conclusion clear enough. Actually, in this sentence, we intended to emphasize the advantages of streamflow-based weights over the weights calculated using climate variables (i.e. temperature and precipitation in this study). As stated in P13, L4-9 when dealing with the raw GCM-simulated streamflows, biases in multi-model mean of annual streamflow are reduced more by the weights based on the impact variable (streamflow), comparing with the weights based on climate variables. Herein, we have modified the expression

of Finding 1 to make this point clearer [P1, L19-21].

*Finding 2: "when applying bias correction to GCM simulations before driving the hydrological model, the climate simulations become rather close to the observed climate, so that compared to equal weighting, the streamflow-based weights do not bring significant differences in the multi-model ensemble mean" P1L21-23: my interpretation is that employing successively two techniques with the same purpose makes the second technique redundant. Reducing the biases in the climate simulations, and then applying MW, makes it extremely difficult for the MW to discriminate between good and poor models. I recognise that BC is applied to the climate simulations and MW to the hydrological simulations, but since all the climate simulations are run through the same hydrological model, calibrated presumably with the forcing dataset also used to perform the BC, the differences in the streamflow simulations are minimal (as shown in Figure 3c and especially 4c). This lack of differences explains why the different weighting methods lead to similar results under current climate (the simulations are almost the same, so how they are combined makes little difference).*

We agree with the referee that in this study, MW loses the ability to discriminate the performances of climate simulations after the bias correction. This is also a finding of this study, which was mentioned in P13, L14-18. We have modified this sentence to make this point clearer.

In fact, MW is not designed for dealing with hydrological simulations but a necessary process to handle the ensemble of multiple climate simulations. Even after bias correction, there still exist some differences between climate simulations. In order to obtain evaluation of climate change impacts, it is unavoidable to choose a MW method to synthesize the simulation results from the ensemble (whether or not bias correction is done). Thus, MW is an indispensable process. Actually, both BC and MW are common procedures in regional impact studies. Although it is common to use equal weighting for bias-corrected ensembles, whether unequal weighting is a better choice remains to be investigated (Alder and Hostetler, 2019). The results of this study show that when the bias correction is done in impact studies, unequal weighting does not bring much difference to the impact evaluation. This supports the usage of equal weighting for bias-corrected ensembles. Nonetheless, we still think that with further development of weighting methods (e.g., more aggressive or multi-objective weighting methods), unequal weighting maybe can help to bring different or more reasonable consequences. The discussion on the weighting methods for the bias-corrected ensembles has been modified in the revised manuscript [P13, L10-27].

*Overall, I suggest shifting the focus from current climatic conditions (for which no climate model and hence MW or BC is necessary) to future conditions (which rely on climate model simulations, which may need BC/MW). In my view, the focus is currently too much on the current conditions. For instance, in the abstract, the authors write "when applying bias correction to GCM simulations before driving the hydrological model, the climate simulations*

*become rather close to the observed climate". This is true because of the nature of bias-correction, and was shown in previous studies (e.g., Hakala et al., 2018). What the grey area in Figures 3d and 4d tells us, however, is that under future conditions, there is substantial spread among the hydrological simulations, although the driving GCM simulations have been bias-corrected (likely because of the different sensitivities of the climate models).*

Thanks for the comment. We agree with the referee that more attention should be paid to the future projections. In the revised manuscript, future simulations are only evaluated in the form of uncertainty (Section 4.4), since there is no observation in the future period to be compared with. In order to partly overcome this problem, we have added the out-of-sample testing in the revised manuscript following the suggestion of referee #2. In out-of-sample testing, the output of each climate model was regarded as the "truth" in turn and the outputs of the remaining 28 climate models were used as simulations to this "truth" model. Then the weights were re-calculated for these remaining models. Since there is a "truth" result for the future period in this case, the performances of weighting methods can be evaluated in terms of reproducing the future "truth".

Figure R1 shows the biases of weighted multi-model mean indices over the Xiangjiang watershed in the out-of-sample testing. The left and right sides of each stick respectively represent the bias at the reference and future periods when one climate model is regarded as truth. **In general, the results of out-of-sample testing are similar to the results using historical observations** (Fig.5). For the streamflow simulated by raw GCM outputs, Fig.R1a-c shows that unequally weighted means more or less become closer to the truth simulation than those of equal weighting in the reference period. The unequal weighting methods can help to reduce the biases in the reference period. Among seven unequal weighting methods, the three methods with the most differentiated weights (REA, UREA and CPI) reduce more biases. In addition, although the biases in the future period tend to be larger than those in the reference period, the weighted means still have a slight improvement in most cases. For the streamflow simulated by bias-corrected GCM outputs (Fig.R1d-f), the multi-model means generalized by unequal weights have similar biases to those of equal weighting. The results of out-of-sample testing over the Manicouagan-5 watershed are shown in Fig.R2, and generally, they are also similar to the results of using observations (Fig.6). The detailed results and analyses of out-of-sample testing has been added in the revised manuscript [P11, L25-P12, L18].

In addition, it is true that when using bias-corrected GCM outputs to simulate streamflows, the differences between ensemble members have been greatly reduced during the reference period while there are still considerable differences in the future period (which had been mentioned in P9, L15-16). This may be because the bias of climate models is nonstationary (Hui et al., 2019; Chen et al., 2015). However, the sentence in the abstract is only an explanation to the results of Finding 2 instead of a focus of this study, but we failed to state this logic clear enough. Therefore, this sentence has been modified to make the focus of this study clearer [P1, L21-23].

*Is there any way to apply MW based on these projected changes, and not based on the streamflow simulations under current climate? In other words, are some of these projections more reliable than others and/or are some projections interdependent, and should be downweighted?*

We thank the referee for this suggestion. Actually, the REA method in this study includes projected future changes when assigning weights. The REA considers both the similarity of a climate simulation to the observation in the reference period and its convergence to the weighted multi-model mean in the future period. Although the weights calculated by the REA method are most differentiated for the bias-corrected ensemble (as Fig. 2 shows), they still bring little impacts on the final results of the multi-model mean. In addition, the PI method considers independency among climate simulations when determining weights, but it only relies on reference values which have been tuned by the bias-correction method. The ability of independent criterion may fail because of the bias correction. This point has been discussed in the revised manuscript [P13, L20-27].

*In summary, my impression is that Finding 1 is relevant but quite foreseeable. I think that Finding 2 is to a great extent due to the experimental design, in particular to the decision to apply BC and MW successively. I encourage the authors to rethink how to best combine MW and BC, for instance by using different periods and/or criteria for the MW.*

We appreciate the comments from the referee. As presented in the last response, the out-of-sample testing has been added in the discussion as a complement. In addition, we have improved our expression that the main focus of this study is to investigate the influences of MW methods on the evaluation of climate change impacts (when the bias correction is or is not done), and to study whether the weighting determined based on the impact variable (streamflow) can induce more reasonable results [P3, L21-28]. This investigation is necessary because MW is a procedure to generalize the results of ensembles and the best way to do it remains questionable. This explanation to the usage of MW and BC has been discussed more clearly in the Discussion section [P13, L10-14].

**References**

[revised manuscript text omitted]

20    **1.4 Upgraded reliability ensemble averaging (UREA)**

Since the REA method may artificially reduce uncertainty by its convergence criterion and only consider one metric (i.e. climatological mean), Xu et al. (2010) proposed upgraded reliability ensemble averaging (UREA) to eliminate the model convergence criterion and to introduce other statistics. Even though multiple climate variables were simultaneously evaluated by multiplying their skill scores in Xu et al. (2010), this study individually evaluated each variable as follows.

$$UREA_i = \left[\frac{\epsilon_a}{abs(B_{a,i})}\right]^{m_1} \times \left[\frac{\epsilon_v}{abs(B_{v,i})}\right]^{m_2} \tag{S6}$$

25    where $B_{a,i}$ and $B_{v,i}$ are the biases of a climate simulation in the average and variance, respectively. $\epsilon_a$ and $\epsilon_v$ represent the natural climate variability in terms of annual average and inter-annual variation, respectively. The variation is measured by the standard deviation for temperature series and by the coefficient of variation for precipitation and runoff series. In addition, if

the absolute value of bias in the average $B_{a,i}$ or variance $B_{v,i}$ is smaller than climate variability $\epsilon$, this climate simulation is regarded to be reliable in the corresponding respect (i.e. $\epsilon_a / \text{abs}(B_{a,i})$ or $\epsilon_v / \text{abs}(B_{v,i})$ is set to 1). 
[revised manuscript text omitted]

---

## Referee Report (RR1)

**Review Wang et al. HESS**

The manuscript by Wang et al investigates the impact of multiple ensemble weighting techniques on the simulations of hydrological impacts for two different river basins. The authors compare the results from a hydrological impact model driven by weighted and unweighted GCM projections. In addition, the authors compare the results from bias-correcting the GCM output before weighting or not. The results show that in case of using raw GCM output to drive the hydrological model a weighted multi-model mean can reduce biases. It is also shown that while different weighting methods behave differently they generally agree on which GCMs should receive higher or lower weights. If bias correction is performed, the weighting method are not beneficial because the bias correction already brings the simulations close to observations. While this is not entirely surprising, the manuscript extensively tests different methods and as far as I know this has not been investigated before. The conclusion that if bias correction is performed before driving the hydrological model, equally weighting each model to calculate the multi-model mean is a viable and conservative choice, will be relevant for other impact modellers.

The manuscript is well written and has been revised carefully. All my comments to an earlier version have been satisfactorily answered and I suggest to publish the manuscript as is.

---

## Author Response (AR2)

Authors' responses to comments

**Does the weighting of climate simulations result in a better quantification of hydrological impacts?**

Hui-Min Wang, Jie Chen, Chong-Yu Xu, Hua Chen, Shenglian Guo, Ping Xie, Xiangquan Li

We appreciate the editor's and the two referees' reviews on the manuscript. These comments are helpful to improve this manuscript. We have carefully studied and responded to all comments point-by-point as follows. For clarity, all comments are given in *italics* and responses are given in plain text. The manuscript has been modified correspondingly.

**Responses to Editor's comments**

We sincerely appreciate the editor for reviewing this manuscript again. We have carefully studied and responded to the comments from the editor and referee #3, since referee #2 has been satisfied with our revision. Please find our specific responses below.

*[Title] I suggest replacing the phrase "a more reasonable" with "a better" or a similar phrase.*

Thanks for the suggestion on the title. We have substituted the phrase "more reasonable" with "better".

*[5, 16-17] In the DT method, are the precipitation and temperature corrected using a cross-correlation, or does each variable is treated independently?*

We are sorry for the unclarity here. In the DT method, each variable is treated independently. In other words, the dependence between precipitation and temperature was not taken into account in this study. Admittedly, it may be more reasonable to use a bias correction method considering the inter-variable dependence. However, our previous study (Chen et al., 2018) showed that the use of a more complicated method does not manifest much advantage over the use of the independent bias correction method for these two watersheds. This has been clarified in the revised manuscript [P5, L22-24].

*[Section 3.2] A figure illustrating the GR4J and CemaNeige models will be useful for the readers. This figure can be presented as Supplementary Material or in the main text.*

Thanks for this helpful suggestion. Figure R1 has been added for illustrating structure of the GR4J-6 hydrological model [Figure S1 in the Supplement].

[Figure]

**Figure R1. The flowchart of the GR4J-6 hydrological model.**

*[Figure 8] Can be presented as Supplementary Material.*

We appreciate the editor's suggestion. This figure has been moved to the Supplement [Figure S5].

*[Section 5] I am missing in the discussion a paragraph about the generalization of the results - what about implications to other climate environments (arid or humid)? urban vs. rural catchments? etc. This should be discussed.*

We agree with the editor that it is necessary to further discuss the implications of results for watersheds in other climate regions.

If weights are determined on climate variables for watersheds in different climate regions, they may manifest different performances on the hydrological impacts. For example, for arid watersheds, whose hydrological regime is more characterized by the intense flow and evaporation, a proper combination for the weights based on temperature and precipitation may be needed in order to obtain a better quantification. For urban watersheds, stormwater contributes to their runoff and weights based on precipitation intensity may be more advantageous. Nevertheless, based on the results of

this study, using impact variables to determine weights may help to circumvent the problem of trade-off and choice of climate variables. However, specific advantages of weights based on impact variables and influences of bias correction on the performances of weighting methods in other types of watersheds still deserve further research.

These points have been added in the Discussion section of the revised manuscript [P15, L4-12].

*[12 20-28] These sentences repeat what is written already in the introduction.*

We agree with the editor that most parts of these sentences repeat the introduction. Here, we intended to state that the results of this study reflect the two problems when unequal weighting methods are used in studies of climate change impacts. Thus, these sentences have been rephrased to curtail the repeating part and to make this point clearer [P12, L21-26].

*[13 10-14] Can be deleted, already mentioned in the introduction.*

Thanks for the suggestion. We have removed these sentences as suggested [P13, L9-10].

*[13 26-27] "Therefore, it is still viable to attend to the bias-corrected ensembles with the equal weighting method" (and similar phrasing in the abstract). This is an awkward way to conclude the results. In fact, the results imply that likely in most cases using bias-correction and equal-weighing is sufficient for hydrological impact studies.*

Thanks for the comment. According to the suggestion, this sentence has been modified as "In hydrological impact studies, it is likely that using equal weighting is viable and sufficient in most cases when bias correction has been applied" [P13, L22-23]. Similarly, the conclusion in the Abstract has also been modified accordingly [P1, L24-25].

*[Figure 9] The values along the y-axis of a-d, b-e, and c-f should be identical to allow comparison between the plots.*

We appreciate the editor's suggestion on this figure. The values along the y-axis have been set to the same for each pair of sub-figures as suggested [Figures 8 and S7].

*[Figure 10] Can be moved to the Supplementary Material.*

Thanks for the suggestion. This figure has been moved to the Supplement [Figure S7].

**Responses to Referee #3's comments**

We sincerely appreciate the referee's comments on the manuscript. We have studied the comments and made relevant corrections to the manuscript. Please find the point-by-point responses below.

*I thank the authors for addressing my comments. I must admit that I find the manuscript and*

*their reply a bit difficult to follow, in particular because of grammatical errors - I apologise in advance if some of the points I raise below are the result of a misunderstanding. I am concerned that future readers will also find this study difficult to read and understand, which is unfortunate at this stage of the review process.*

We would like to thank the referee for reviewing our manuscript. We feel sorry that some grammatical errors in the last reply led to some confusions. We have carefully reviewed the manuscript and fixed the grammatical errors to make our statements clearer. Please find our specific responses below.

*In the abstract, the authors now state:*

*1. "when using raw GCM outputs, streamflow-based weights better represent the mean hydrograph and reduce more biases of annual streamflow than the weights calculated using climate variables". This is an interesting and potentially impactful result. But where is it shown? I cannot find a Figure or Table supporting the above statement. Table 3 shows the entropy of weights computed using temperature and precipitation, and Figure S1 the weights based on temperature and precipitation, but as I understand it, this does not demonstrate the benefits of computing weights using streamflow instead of temperature/precipitation. In their reply to my comment, the authors refer to P13, L14-18, which is part of the discussion and does not refer to any Figures or Tables in the manuscript. Please clarify. Please also note that the above sentence is not grammatically correct.*

Sorry for the unclear sentence. In our last response, the citation to "*P13, L14-18*" was used to demonstrate that the results of this study confirm the point of the referee (i.e. "*Reducing the biases in the climate simulations, and then applying MW, makes it extremely difficult for the MW to discriminate between good and poor models*"), and it did not serve to show results for the experiment of using raw GCM outputs to simulate streamflow.

We feel sorry that we did not clearly point out relevant results in our last response. Actually, better performances of streamflow-based weights on mean hydrograph are presented in Figures 3, 4 & S3 for both watersheds and described in Section 4.2 *Impacts on the hydrological regime* (especially in P9, L4-10). Fewer biases of annual streamflow are presented in Figures 5, 6 & S4 and described in Section 4.3 *Bias in multi-model mean* (especially in P10, L9-11 & 24-27). Also, the advantages of streamflow-based weights were also discussed in the first paragraph of the Discussion section. Herein, we have added citation to relevant figures in this paragraph to make this analysis clearer [P12, L26-P13, L8].

In addition, in order to fix grammatical errors, we have corrected the sentence in the abstract as "when using raw GCM outputs to simulate streamflows, streamflow-based weights have a better performance in reproducing observed mean hydrograph than climate-variable-based weights" [P1,

L19-21].

2. *"when applying bias correction to GCM simulations before driving the hydrological model, the streamflow-based unequal weights do not bring significant differences in the multi-model ensemble mean and uncertainty of hydrological impacts, since bias-corrected climate simulations become rather close to observations."* As also stressed by reviewer 2, this is expected, as by construction, bias-correction forces climate simulations to look like observations, i.e. artificially reduces differences between them (e.g., Hakala et al., 2018), thereby making it difficult to differentiate between good and poor models. Hence, on its own, this result does not justify publication in my view.

We appreciate the referee's comment on this conclusion. Sorry that we failed to make this point clear enough in the previous version. As stated in P8, L15-16 & P9, L11-15, we agree that in the experiment of using bias-corrected GCM outputs to simulate streamflows, the ability of weighting methods to differentiate performances among different GCMs is affected by the bias correction, which is also observed in Hakala et al. (2018). This is the reason why the results of unequal weighting are similar to those of equal weighting in this experiment. This is also discussed in the Discussion section (P13, L7-13). However, this does not mean that the performances of model weighting methods in this case do not deserve research. Although the equal weighting is often used for bias-corrected ensembles by default in many regional hydrological impact studies, whether this method is viable or sufficient remains unclear. This study focuses on this problem and can offer a reference for the use of weighting methods in hydrological impact studies. In addition, bias correction methods are usually applied to climate variables for hydrological impacts studies. Even though most of bias correction methods can reduce the bias of climate model simulations in terms of a few statistical metrics, no bias correction methods are perfect to remove all the biases. However, hydrological simulations reflect the overall performance of climate simulations, and small biases in climate simulations (in terms of a few metrics) may result in large biases in hydrological simulations, especially taking into account the fact that the climate to hydrological process is nonlinear. Thus, if unequal weighting methods consider criteria that are different to the bias correction, they may have potentials to induce better quantification. This problem deserves further studies.

These points have been clarified in the revised manuscript [P12, L19-21 & P13, L22-29].

The authors conducted additional analysis based on a pseudo-reality experiment to explore the consequence of model weighting under future climate. I thank them for their effort. The results are consistent with those based on current climatic conditions: bias-correction makes it very difficult to distinguish between good and poor models, and there are essentially no benefits to implement a weighting scheme, as the weighting schemes considered do not reduce biases more than an equal weight strategy (see Figures 9d-f and Figures 10d-f). Again, I believe that sequentially applying bias-correction and model weighting based on bias-corrected

*simulations is a flawed approach.*

Thanks for the comments on the out-of-sample testing. We feel that we need to explain better our motivation for conducting this study. On the one hand, we agree that unequal weighting methods do not bring significant improvements on the multi-model mean compared to the equal weighting when using bias-corrected ensembles. On the other hand, we think that model weighting should not be regarded as a supplementary process but a necessary process in impact studies. No matter whether bias correction is done before driving the hydrological model or not, a decision on the weighting methods is always necessary in order to obtain multi-model mean or uncertainty evaluation. Albeit the normal and default choice of equal weighting, whether this choice is viable and sufficient remains unclear, and this study focuses on this question. We have clarified this point in the Discussion section of the revised manuscript [P12, L19-21].

*Furthermore, I am concerned that the results of this study might be misinterpreted. Readers might believe that there are no benefits to use weighting when the climate simulations have been bias-corrected. It is possible, however, that there are benefits of combining bias-correction and model weighting, but I would argue that the model weighting should be done using other criteria than those considered for bias-correction. For instance, the model weighting could reduce the influence of (or exclude) models with erroneous behaviour (for instance climate models creating snow towers, as it is the case for EURO-CORDEX members) or are unable to capture key processes such as atmospheric rivers or atmospheric patterns (e.g., NAO). This could potentially constrain the ensemble (e.g., Padrón et al., 2018) and could be a successful application of model weighting and bias correction. But this would require substantial additional analysis.*

Thanks for the comment. We are sorry that we might fail to explain our motivation and main findings clear enough in the previous version. This study does not mean to demonstrate that unequal weighting is totally not beneficial for bias-corrected ensembles. As stated in the P15, L27-29, this study concludes that equal weighting method is not perfect but a viable and conservative choice so far to deal with bias-corrected GCMs in hydrological impact studies.

In addition, this study used two weighting methods (REA and PI) which consider criteria that are different from the bias correction. However, both methods still do not bring differences to the final results. This has been discussed in P13, L15-20. Actually, as stated in the responses above, we agree with the referee that future development of model weighting may have potentials to induce better quantification of hydrological impacts, and our study can offer a reference for this. As stated in P14, L13-14, considering dynamic reliability of GCM simulations in model weighting may help to induce a better quantification of hydrological impacts. In addition, considering criteria different to bias correction methods in the model weighting may also help to induce a better quantification of hydrological impacts.

This has been clarified in the revised manuscript [P13, L27-29 & P14, L13-14].

> *Overall, I think that the paper lacks a clear vision and a clear message. In their title, the authors ask "Does the weighting of climate simulations result in a more reasonable quantification of hydrological impacts?" But then they do not clarify what they mean by "reasonable"...*

We appreciate the referee's the comment on the title. First of all, we have changed the phrase "a more reasonable" to "a better" as suggested by the editor. In addition, following many studies in model weighting, "better" in the title means a multi-model ensemble mean which is more similar to the observation and less uncertainty in the ensemble. Both objectives have been studied in this work. Accordingly, this point has been stated in the Discussion section [P12, L19-21].

$$UREA_i = \left[\frac{\in_a}{abs(B_{a,i})}\right]^{m_1} \times \left[\frac{\in_v}{abs(B_{v,i})}\right]^{m_2} \tag{S6}$$

where $B_{a,i}$ and $B_{v,i}$ are the biases of a climate simulation in the average and variance, respectively. $\in_a$ and $\in_v$ represent the natural climate variability in terms of annual average and inter-annual variation, respectively. The variation is measured by the standard deviation for temperature series and by the coefficient of variation for precipitation and runoff series. In addition, if

the absolute value of bias in the average $B_{a,i}$ or variance $B_{v,i}$ is smaller than climate variability $\epsilon$, this climate simulation is regarded to be reliable in the corresponding respect (i.e. $\epsilon_a/\operatorname{abs}(B_{a,i})$ or $\epsilon_v/\operatorname{abs}(B_{v,i})$ is set to 1). The parameters $m_1$ and $m_2$ represent the weight assigned to two metrics and are both set to 1 in this study.

**S1.5 Bayesian model averaging (BMA)**

5      Bayesian model averaging (BMA) is a statistical inference approach to obtain probabilistic forecasts from multi-model ensemble simulations based on Bayes theory. BMA has been used to develop probabilistic predictions for ensembles of weather forecasting models, climate models or hydrological predictions (Duan et al., 2007; Min et al., 2007; Raftery et al., 2005). Denote $y$ as the variable to be predicted, $D = [y_1^o, y_2^o, \ldots, y_T^o]$ as the observed series with a length of $T$, and $f = [f_1, f_2, \ldots, f_N]$ as the ensemble of series simulated by climate models. Based on the total probability rule, the probability density function of

10    the prediction $p(y|D)$ can be presented as follows.

$$p(y|D) = \sum_{i=1}^{N} p(f_i|D) \cdot p_i(y|f_i, D) \tag{S7}$$

where each simulation $f_i$ is associated with a conditional probability density function, $p_i(y|f_i, D)$, which represents the conditional distribution of $y$ on $f_i$, given that $f_i$ is regarded as the best simulation for $D$. The posterior probability $p(f_i|D)$ represents the likelihood that a simulation is the right simulation. It can also be seen as the weight, $w_i = p_i(y|f_i, D)$, which reflects the capability of a simulation to reproduce the observation. Then, the posterior mean is as follows.

[revised manuscript text omitted]